

# Spatio-temporal information propagation using sparse observations in hyper-resolution ensemble-based snow data assimilation

Esteban Alonso-González[1], Kristoffer Aalstad[2], Norbert Pirk[2], Marco Mazzolini[2], Désirée Treichler[2], Paul Leclercq[2], Sebastian Westermann[2], Juan Ignacio López-Moreno[3], and Simon Gascoin[1]

[1]Centre d'Etudes Spatiales de la Biosphère, Université de Toulouse,CNRS/CNES/IRD/INRAE/UPS, Toulouse, France
[2]Department of Geosciences, University of Oslo, Oslo, Norway
[3]Instituto Pirenaico de Ecología, CSIC, Zaragoza, Spain

**Correspondence:** Esteban Alonso-González (esteban.alonso-gonzalez@univ-tlse3.fr)

**Abstract.** Monitoring the snowpack remains challenging in part due to the limited availability of observations. On the one hand, the deployment of dense ground-based monitoring networks is hampered by logistical hurdles. On the other hand, satellite-based remote sensing products provide only partial information about the snowpack, often limited to snow-covered area or surface temperature. Numerical models are a valuable tool to help fill the gaps in snowpack monitoring. Model performance
is nonetheless contingent upon the quality of meteorological forcing, which is often highly uncertain especially in complex terrain. To address these limitations, data assimilation techniques that integrate available observations with snow models have been proposed as a viable option to simultaneously help constrain model uncertainty and add value to observations by improving estimates of the snowpack state. However, the propagation of information from spatially sparse observations in high resolution simulations remains an under-explored topic. To remedy this, the development of data assimilation techniques that
can spread information in space is a crucial step. Herein, we examine the potential of spatio-temporal data assimilation for integrating sparse snow depth observations with hyper-resolution (5 m) snow simulations in the Izas central Pyrenean experimental catchment (Spain). Our experiments were developed using the Multiple Snow Data Assimilation System (MuSA) with new improvements to tackle the spatio-temporal data assimilation. Therein, we used a Deterministic Ensemble Smoother with Multiple Data Assimilation (DES-MDA) with domain localization.

Three different experiments were performed to showcase capabilities of spatio-temporal information transfer in hyper-resolution snow simulations. Experiment I employed the conventional geographical Euclidean distance to map the similarity between cells. Experiment II utilized the Mahalanobis distance in a multi-dimensional topographic space using terrain parameters extracted from a digital elevation model. Experiment III utilized a more direct mapping of snowpack similarity from a single complete snow depth map together with the easting and northing coordinates. Although all experiments showed a noticeable
improvement in the snow patterns in the catchment compared with the deterministic open loop in terms of correlation ($r = 0.13$) and root-mean-square error (RMSE$= 1.11$ m), the use of topographical dimensions (Experiment II, $r = 0.63$ and RMSE$= 0.89$ m) and observations (Experiments III, $r = 0.92$ and RMSE$= 0.44$ m) largely outperform the simulated patterns in Experiment I ($r = 0.38$ and RMSE$= 1.16$ m). At the same time, Experiments II & III are considerably more challenging to set up. The results of these experiments can help pave the way for the creation of snow reanalysis and forecasting tools that
can seamlessly integrate sparse information from national monitoring networks and high-resolution satellite information.



# 1 Introduction

Covering nearly half the northern hemisphere's land surface every year (Déry and Brown, 2007), the snowpack is a major component of the terrestrial cryosphere. Its reflective, insulating, and large water storage capacity makes the snowpack a key modulator of biogeophysical and biogeochemical processes in the areas where it is present (Zhang, 2005; DeWalle and Rango, 2008). It controls the ecology of cold regions (Slatyer et al., 2022; Pirk et al., 2023), and provides key freshwater resources to snow dominated areas and large downstream regions (Mankin et al., 2015). The mass and duration of the global snow cover has been reduced by recent climate change (Pulliainen et al., 2020), and it is anticipated that these impacts will continue in the coming decades (Mudryk et al., 2020). All this highlights the increasing need for monitoring the snowpack, both to evaluate the impacts of climate change on the cryosphere and to design better adaptation strategies (Sturm et al., 2017).

Snowpack monitoring is a difficult task, particularly in remote regions where meteorological conditions are often severe and logistics are challenging (Fayad et al., 2017). The snowpack exhibits a significant degree of spatial variability (López-Moreno et al., 2013), especially in areas of complex topography, as a result of processes such as preferential deposition, wind redistribution, and avalanching (Comola et al., 2019; Vionnet et al., 2021). This complexity calls into question the representativeness of the typically sparse point scale snow measurements from automatic weather stations as well as those performed during manual field campaigns. For these reasons, numerical snowpack simulations are widely used for both scientific and operational goals. Using physically-based energy and mass balance models it is possible to continuously (i.e., without gaps in space and time) simulate the state of the snowpack represented through a number of variables that are currently difficult to estimate directly from observations. This includes the snow water equivalent (SWE), which is of great interest for hydrologists as it constitutes a direct estimate of the freshwater resources stored in the snowpack. Unfortunately, numerical snowpack models require accurate high spatio-temporal resolution meteorological forcing, which is often difficult to obtain. The uncertainties in the forcing propagate through the snowpack numerical models inducing strong biases in the simulations. In fact, snowpack simulation errors tend to mostly originate from the uncertainty in the forcing, rather than a lack of knowledge concerning snow physics (Raleigh et al., 2016). Apart from the uncertainty induced by the forcing, redistribution processes may cause strong differences between simulations and reality. Despite recent model developments, explicitly accounting for wind and avalanche driven snow redistribution remains challenging and introduces yet further degrees of freedom in the modeling exercise.

Earth observation satellites provide information on various snow-related variables such as the snow cover extent, snow depth, snow surface temperature or albedo. However, remote sensing observations are limited by revisit times, spatial resolution, cloud obstruction, viewing geometry, and spectral resolution (Ju and Roy, 2008; Dozier et al., 2008). As a result, it is not yet possible to generate spatially continuous maps of these variables with a sufficient temporal resolution in near real time (e.g. daily). In addition, remote sensing does not allow the retrieval of key snow variables like the snow water equivalent (SWE) in mountain regions (Dozier et al., 2016).

Data assimilation (DA) has emerged as a promising method for enhancing uncertain numerical snowpack simulation results using available in-situ or remotely sensed observations (Largeron et al., 2020). The use of DA methods allows for the correction of errors in meteorological forcing, resulting in improved predictions of snow models through the incorporation of the infor-



mation distilled from observations while taking into account their associated uncertainties. In particular, the assimilation of in situ and remote sensing products have already shown to improve snow simulations, including in situ snow depth (Smyth et al., 2019), fractional snow-covered area (Aalstad et al., 2018), land surface temperature (Alonso-González et al., 2022b), albedo (Kumar et al., 2020), or even assimilating satellite reflectances directly (Revuelto et al., 2021b). If the objective is to reconstruct the SWE distribution in the absence of direct SWE observations, snow depth is likely the most useful variable since it explains most of the SWE variability in many regions (Sturm et al., 2010). Assimilating snow depth also enables one to benefit from the wealth of in situ snow depth measurements from automatic stations or manual sampling (Smyth et al., 2019) or from emerging satellite products (Marti et al., 2016; Lievens et al., 2022; Deschamps-Berger et al., 2022). Unfortunately, the assimilation of in situ or remotely sensed snow depth poses several challenges. One major challenge is the mismatch in scale between the sparse spatial sampling of in situ or altimetry-based snow depth measurements and the gridded or semi-distributed geometry of the simulations (Molotch and Bales, 2005). Even imagery-based remote sensing approaches to retrieve snow depth can present considerable spatial gaps due to orbital constraints, sensor swath or the presence of clouds in the image. This results in a vast number of model grid cells that lack local observations, requiring the propagation of information from observed to unobserved areas within snow DA frameworks to be able to fully exploit existing monitoring systems.

Despite the above issues, the question of spatial information transfer has thus far received relatively little attention from land surface modelers in general (De Lannoy et al., 2022) and snow modelers in particular. Spatio-temporal DA, also known as "3D" DA, has the benefit of propagating information from observations both in time and space with the potential to fill gaps in otherwise sparse observations (Reichle and Koster, 2003). At the same time, unlike purely temporal (i.e., one dimensional) DA, defined here as the case where each spatial unit is treated independently, adding spatial dimensions vastly increases the dimensionality of the underlying Bayesian inference problem when performing a global (domain-wide) analysis. This makes spatio-temporal DA more challenging to implement in practice than temporal DA. For example, with ensemble-based DA methods a global analysis would often require an intractable exponential increase in ensemble size to avoid degeneracy (Bocquet et al., 2017; Farchi and Bocquet, 2018).

Thanks to developments fueled by operational numerical weather prediction (Houtekamer and Zhang, 2016; Bannister, 2017), tailor-made methods exist that make ensemble-based DA feasible even for extremely high-dimensional spatio-temporal problems with on the order of a billion state variables. These are known as localization methods (Sakov and Bertino, 2011; Chen and Oliver, 2017) which can be split in two distinct types: covariance localization and domain localization (also know as local analysis). Both localization methods effectively alleviate computational issues by limiting the radius of influence of observations, thus reducing the spatio-temporal dimensionality of the DA problem. These methods are often applied as a remedy to spurious correlations that can cause unphysical or extreme long-range information transfer from observations. Spurious correlations are, however, only a symptom of the limited ensemble size in high-dimensional problems which can lead to a deficiency of the low rank approximation of ensemble Kalman methods (Sakov and Bertino, 2011; Evensen et al., 2022) and make particle methods degenerate (Farchi and Bocquet, 2018). Given an infinite ensemble size these issues would disappear, but for all practical purposes they remain a major concern. This makes localization indispensable for designing functioning high-dimensional spatio-temporal DA systems.



Despite receiving considerably less attention than its temporal counterpart, some examples of spatio-temporal snow DA of can be found. De Lannoy et al. (2010) investigated spatio-temporal assimilation of synthetic coarse-scale (25 km) passive microwave SWE retrievals in high resolution (1 km) simulations of the Noah land surface model using the ensemble Kalman filter (EnKF) with covariance localization based on horizontal distance. In a follow up study, De Lannoy et al. (2012) performed similar spatio-temporal experiments but using real data, showing the added value of jointly assimilating passive microwave

and optical retrievals at different resolutions. Magnusson et al. (2014) performed spatio-temporal assimilation experiments using the EnKF and Optimal Interpolation with three dimensional (horizontal and vertical) localization, effectively transferring information from ground-based point SWE observations into a distributed snow model over Switzerland. More recent studies have suggested different strategies using particle filters (PF) for spatio-temporal snow DA. Cantet et al. (2019) adopted a heuristic approach to propagate information from sparse SWE observations over the Canadian province of Quebec using a PF

with spatio-temporally correlated perturbations by performing isolated analyses for grid cells where observations were available and subsequently spatially interpolating the posterior weights. This methodology was further developed by Odry et al. (2022) who suggested using the Schaake shuffle method (Clark et al., 2004) in order to alleviate spatial discontinuities that can arise when resampling. Cluzet et al. (2021) suggested alternative promising approaches to spatio-temporal snow DA using a PF in synthetic experiments with a semi-distributed geometry by combining a form of domain localization with observation error

inflation. Cluzet et al. (2022) extended these approaches using data from a real snow depth observation network over the French Alps and Pyrenees, showing marked improvements compared to both the open-loop and the current operational approach of Météo-France.

The aforementioned spatio-temporal snow DA studies were typically performed at moderate resolution, in semi-distributed geometries, and/or using relatively simple snow models. In addition, the quantification of the spatial relationships between

cells was typically derived from the Euclidean distance. Including a measure of the elevation proximity between cells helped to account for large differences in SWE for cells that were close in the horizontal dimension but located at different elevations (Magnusson et al., 2014). However, at hyper spatial resolution (e.g. 5 m), the behavior of the snowpack is also correlated with other topographic variables like slope, aspect, etc. (López-Moreno et al., 2017; Elder et al., 1991; Revuelto et al., 2020), that modulate key processes such as incoming radiation (Liston and Elder, 2006) and snow redistribution by wind drift(Vionnet

et al., 2021; Sharma et al., 2023). Hence, such topographical characteristics should also be considered in the DA process. It is thus necessary to define generalized distances in higher dimensional spaces. In the context of the ongoing proliferation of high resolution satellite data and with the objective of maximizing this data together with the benefits of point scale observations in snow DA, it is imperative to push the current limits of these techniques so as to improve fine scale snow simulations. In this paper we explore the potential of sparse snow depth observations to update hyper-resolution snowpack models using

topographical variables in addition to the usual spatial dimensions so as to maximize the contribution of observations to the analysis. We have implemented this new spatio-temporal information propagation capability such that it can be applied to a plethora of emerging snow DA scenarios. We present the results from three experiments based on real (rather than synthetic) data that explore increasingly sophisticated approaches to information propagation in hyper-resolution snow DA.



## 2 Data and methods

### 2.1 Observations and meteorological forcing

This work is based on the data available from a time series of 12 hyper-resolution (5 m spatial resolution) snow depth maps collected during a single snow season over the Izas experimental catchment in the Pyrenees (55 hectares Revuelto et al., 2017) (Fig. 1). The maps were retrieved from fixed wing drone surveys using structure from motion techniques. The drone surveys were conducted in 2020 on 14/Jan, 03/Feb, 24/Feb, 11/Mar, 29/Apr, 3/May, 12/May, 19/May, 26/May, 02/Jun, 10/Jun, 21/Jun. Several surveys were conducted before peak accumulation (estimated to be in mid March) but the late snowmelt season in May and June was sampled most densely. This dataset has been extensively validated (Revuelto et al., 2021a, c) and ingested in purely temporal snow DA experiments in a previous study (Alonso-González et al., 2022a). According to these previous experiments, the error variance of the observations was assumed to be $\sigma_y^2 = 0.04 \, \mathrm{m}^2$.

We used these maps to generate sparse observations to be assimilated by randomly selecting 20 cells among all the available grid cells for every map. The complete maps were also used to evaluate the posterior simulations. We assumed that the snow depth maps were an independent source of evaluation given that the assimilated observations only represent $0.11\%$ of the $18\,442$ simulated grid cells. The random draw of 20 cells was performed independently for each map, emulating a real case where an observer makes sporadic snow depth probe measurements throughout the snow season. The random sampling led to the selection of several snow-free cells, because many snow surveys were conducted late in the snow season (Figure 1).

We used a meteorological forcing dataset that was previously generated by Alonso-González et al. (2022a) using the MicroMet meteorological distribution system (Liston and Elder, 2006). MicroMet was applied to downscale hourly outputs of the ERA5 atmospheric reanalysis from their native resolution of $0.25°$ (Hersbach et al., 2020) to the same 5 m grid as the drone observations.

This meteorological forcing was used to drive the Multiple Snow data Assimilation system (MuSA, Alonso-González et al., 2022a). MuSA is an open-source snow data assimilation toolbox designed to assimilate various observations of the snowpack into an ensemble of snowpack simulations generated by the energy and mass balance model the Flexible Snow Model (FSM2 Essery, 2015). It was necessary to include several modifications in the MuSA code, as detailed in the Section 2.4. All the experiments where performed using the most complex parameterization scheme of FSM2, as its the default configuration of MuSA. In MuSA, FSM2 is run in a distributed fashion but in its current version the wind redistribution is not simulated. We expect the spatio-temporal DA to be able to mimic the wind redistribution process implicitly by correcting the precipitation scaling parameters.

### 2.2 Spatio-temporal data assimilation

DA is a term used in the geosciences for the ubiquitous exercise of combining models with observations (Carrassi et al., 2018). This exercise often becomes quite a formidable computational challenge to implement in practical DA when complex mechanistic models and real observations are involved. Several approaches have been developed to deal with the computational challenge of combining models with observations (Evensen et al., 2022; Murphy, 2023). These can broadly be divided into



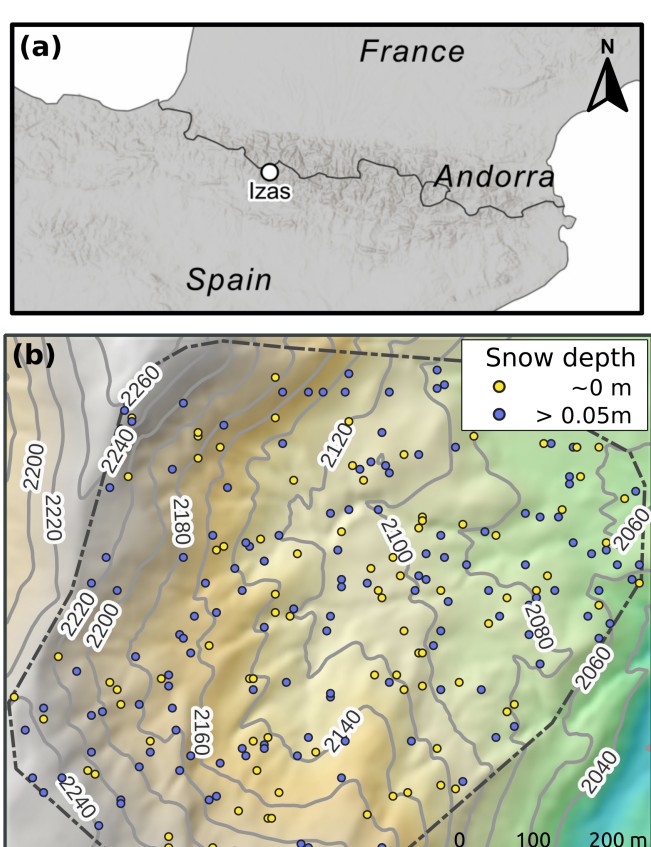

**Figure 1.** Location of the Izas Experimental catchment in the Spanish Central Pyrenees (a). Topography of the catchment derived from a summer drone survey, and a hillshade map showing cells (circles) where observations were obtained from the random sampling strategy during the observational period in 2020. The colour indicates which of the cells had snow cover (defined as snow depth over 5 cm) at the time of the observation (b).

Monte Carlo and variational techniques. Since the latter involves the computation of gradient terms (Bannister, 2017) that can be non-trivial, it has so far been rarely used for snow DA. As such, Monte Carlo techniques (Chopin and Papaspiliopoulos, 2020) are widely used in snow DA, with the most common approaches being ensemble Kalman and particle methods (see

Alonso-González et al., 2022a, and references therein). The application of particle methods to high-dimensional problems, such as those that arise in spatio-temporal settings, remains an outstanding challenge at the research frontier of DA due to particle degeneracy that occurs in the absence of robust particle localization methods (Farchi and Bocquet, 2018).

In this work we focus on ensemble Kalman methods which lend themselves well to spatio-temporal DA thanks to their Gaussian properties and the compatibility with localization methods, as is clear from existing practices in the broader DA

community (Sakov and Bertino, 2011; Chen and Oliver, 2017; Evensen et al., 2022). Indeed, ensemble Kalman methods are



closely related to many spatial modeling techniques, including kriging (Krige, 1951; Matheron, 1963) methods in geostatistics (Bertino et al., 2003; Chilès and Delfiner, 2012) and the nearly equivalent Optimal Interpolation (Eliassen, 1954; Gandin, 1963) methods that were widely (and are still occasionally) used in operational DA (Talagrand, 1997). What sets the ensemble Kalman methods apart from these methods is the use of a nonlinear mechanistic dynamical model for probabilistic prediction.

The intimate relationship between these methods is not surprising given that they can be viewed as special cases of a more general mathematical concept known as a Gaussian process (Rasmussen and Williams, 2005) that is widely used both in statistics (Cressie and Wikle, 2011; Wikle et al., 2019) and machine learning (MacKay, 2003; Murphy, 2022, 2023).

In spatio-temporal DA, information from observations can be spread across multiple grid cells through non-local observations, correlated observation error, or prior dependence (van Leeuwen, 2019). Non-local observations are observations that can

not be confined to a single model grid cell, as is the case when the resolution of the observations is coarser than that of the model (e.g. De Lannoy et al., 2012). Although it is important to correctly account for the effects of such non-local observations in DA (van Leeuwen, 2019), they are not pertinent to the case of of local sparse observations considered in this study. Moreover, non-local observations only affect the grid cells that their support overlaps with and thus do not help resolving the challenge of transferring information from observed to unobserved locations. Not accounting for correlated observation errors,

which occur frequently in satellite remote sensing, can also degrade the performance of the assimilation (Carrassi et al., 2018). In this study, based on our knowledge of the drone observations outlined in Revuelto et al. (2021a, c), we make the common assumption that the observation errors are uncorrelated by assigning a diagonal observation error covariance matrix of the form $\mathbf{R} = \sigma_y^2 \mathbf{I}$. Moreover, despite their potential importance, including observation error correlations would not in itself allow for information transfer from observed to unobserved locations. This leaves prior dependence as the sole mechanism that permits

information to be transferred from spare local observations to unobserved locations. It is this mechanism, explained in detail in Appendix A, that we exploit in this study.

### 2.2.1 Ensemble generation

Monte Carlo methods in general and ensemble Kalman methods in particular require the use of an ensemble (i.e. a collection) of model realizations. To generate the prior ensemble of simulations, we used time invariant (within a water year) prior parameter

ensemble to perturb the precipitation (multiplicative) and air temperature (additive) forcing variables. To bound these parameters within certain limits while satisfying the Gaussian prior assumption of ensemble Kalman methods they were drawn from logit-normal distributions as outlined in (Aalstad et al., 2018). The parameters are updated with the ensemble Kalman analysis step in the transformed (Gaussian) space but fed through the forward model in the physical (untransformed) space (Alonso-González et al., 2022a). The mean $\mu_0$ and standard deviation $\sigma_0$ of the underlying normal distributions hyper-parameters,

control the shape of our weakly informative prior probability distributions (Banner et al., 2020). These hyper-parameters were selected based on conservative expectations of the range of uncertainty in the meteorological forcing, which we distilled from experience obtained in previous studies at Izas. For temperature, the prior additive perturbation parameters were drawn from a logit-normal distribution bounded between $-8$ and $8$ K, with hyperparameters $\mu_0 = 0$ and $\sigma_0 = 0.5$. The prior multiplicative





perturbation parameters for precipitation were drawn from a logit-normal distribution bounded between $0$ and $8$ with with $\mu_0 = -1.6$ and $\sigma_0 = 1$. The number of ensemble members was fixed at $N_e = 100$ for all experiments.


Each of the transformed prior perturbation parameters were drawn from independent high-dimensional multivariate normal distributions (i.e., in the transformed space) by constructing prior spatial covariance matrices. This prior dependence structure allows for associations between the parameters in all the grid cells in our domain which is key for information propagation as outlined in Appendix A. For simplicity, the temperature and precipitation perturbation parameters were assumed to be indepen-

dent but it would also be possible to consider a correlation between parameter types. To generate the prior spatial covariance matrices we first used the $5^{th}$-order piecewise rational function proposed by Gaspari and Cohn (1999) (their Eq. 4.10), henceforth referred to as GC (Fig. 2). The GC functions is widely used for localization (Sakov and Bertino, 2011; Chen and Oliver, 2017) to generate a prior correlation matrix based on the the pairwise distances between grid cells in the simulation:

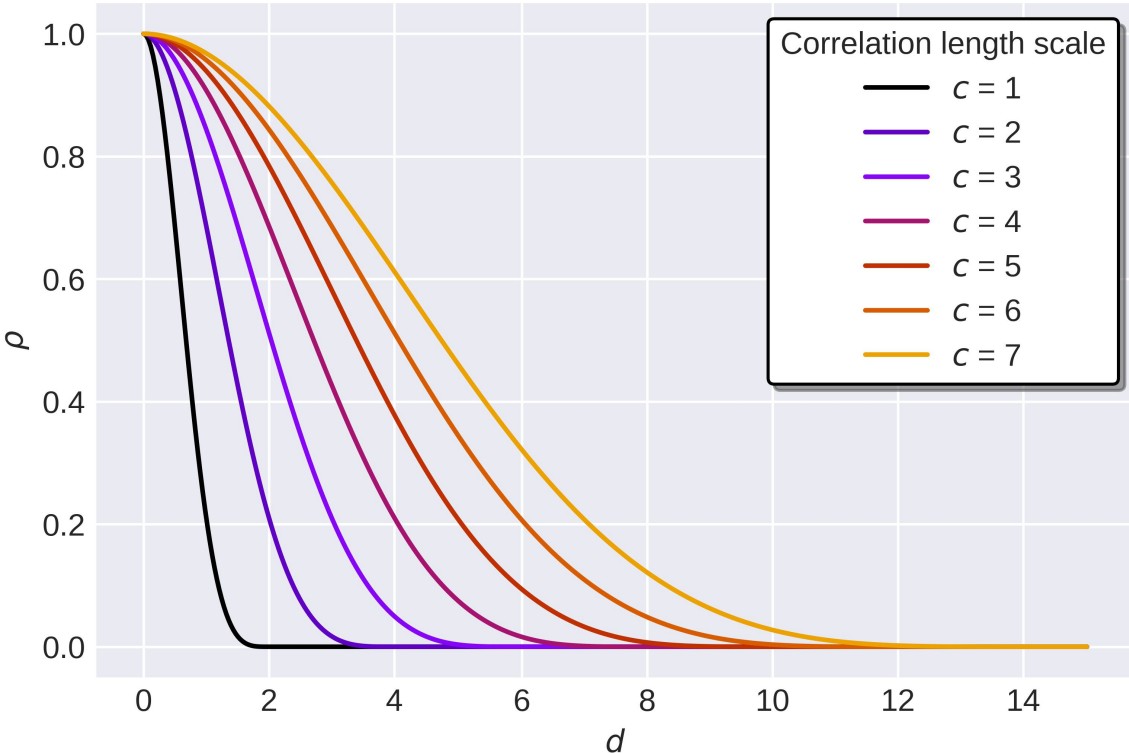

**Figure 2.** Example of correlation values ($\rho$) estimated from distances ($d$) by the Gaspari and Cohn function for different correlation length scales.





$$\rho(d,c) = \begin{cases} -\frac{1}{4}\left(\frac{d}{c}\right)^5 + \frac{1}{2}\left(\frac{d}{c}\right)^4 + \frac{5}{8}\left(\frac{d}{c}\right)^3 - \frac{5}{3}\left(\frac{d}{c}\right)^2 + 1, & \text{for } 0 \leq \left(\frac{d}{c}\right) \leq 1, \\ \frac{1}{12}\left(\frac{d}{c}\right)^5 - \frac{1}{2}\left(\frac{d}{c}\right)^4 + \frac{5}{8}\left(\frac{d}{c}\right)^3 + \frac{5}{3}\left(\frac{d}{c}\right)^2 - 5\left(\frac{d}{c}\right) + 4 - \frac{2}{3}\left(\frac{d}{c}\right)^{-1}, & \text{for } 1 \leq \frac{d}{c} \leq 2, \\ 0, & \text{for } \frac{d}{c} > 2. \end{cases} \tag{1}$$

where $d/c$ is a normalization of the distance $d$ between a pair of grid cells by the correlation length scale $c$. With this GC function, the correlation drops off rapidly from 1 at $d = 0$ to 0.5 at $d \simeq 0.68c$ and finally to zero at $d = 2c$. The correlation length scale $c$ thus plays an important role as a hyperparameter that should be tuned in a sensitivity analysis or determined empirically from data using geostatistical methods (Chilès and Delfiner, 2012). In DA, the distance $d$ in the GC function is typically the Euclidean distance between two spatial grid cells defined in two (easting and northing) or three (with elevation)

dimensional geographic space. The concept can be generalized, however, to be any measure of distance between two grid cells as detailed in Section 2.3. In our experiments, we have selected two different values of the hyperparameter $c$ after manual tuning. For Experiment I $c = 100$, while for Experiments II and III $c = 5$ (Section 2.3). The difference in the magnitude of the $c$ value in the different experiments is a consequence of the covariance-based normalization of the distance matrix $(d_{ij})$ of Experiments II and III.

Based on the correlations calculated from the GC function, we construct a correlation matrix $\boldsymbol{\rho}$ with dimensions $N_g \times N_g$ where $N_g$ is the total number of grid cells in our domain. This matrix contains the prior spatial correlations for a particular (transformed) perturbation parameter. In general both the choice of distance $d$ and cutoff $c$ can differ between perturbation parameters, but for simplicity here we kept them the same for both the temperature and precipitation perturbation parameters. Through the close relationship between correlation and covariance, it is then possible to construct a prior covariance matrix $\mathbf{C}_0$

by also taking into account the column vectors $\boldsymbol{\sigma}_0$ that contain the prior standard deviations of the transformed perturbation parameters across the domain through

$$\mathbf{C}_0 = \boldsymbol{\rho} \odot \left(\boldsymbol{\sigma}_0 \boldsymbol{\sigma}_0^{\mathrm{T}}\right), \tag{2}$$

where $\odot$ is the element-wise product and $(\cdot)^{\mathrm{T}}$ denotes the transpose. This is a general expression which happens to simplify in our case since we assume that the prior standard deviations are constant in space, such that $\boldsymbol{\sigma}_0 \boldsymbol{\sigma}_0^T$ is just a constant matrix

containing the prior variance. Given that $\mathbf{C}_0$ is positive definite, we can obtain its matrix square root via Cholesky factorization $\mathbf{C}_0 = \mathbf{L}_0 \mathbf{L}_0^{\mathrm{T}}$ allowing us to sample an $N_g \times N_e$ matrix $\mathbf{U}_0$ of prior ensemble of transformed perturbation parameters $\mathbf{U}_0$ from the multivariate Gaussian prior $\mathcal{N}(\boldsymbol{\mu}_0, \mathbf{C}_0)$ through (Murphy, 2022)

$$\mathbf{U}_0 = \boldsymbol{\mu}_0 \mathbf{1}_{N_e}^{\mathrm{T}} + \mathbf{L}_0 \boldsymbol{\zeta}, \tag{3}$$

where $\boldsymbol{\zeta} \sim \mathcal{N}(0,1)$ is a $N_g \times N_e$ matrix of pseudo-random draws from a standard normal distribution and $\mathbf{1}_{N_e}$ is a $N_e \times 1$ vector

of ones. We repeat this sampling for each of the perturbation parameters, effectively drawing both spatial prior ensembles independently and then combine them to form a joint prior ensemble matrix $\mathbf{U}_0$ with $N_p \times N_g$ rows (where in our case we have $N_p = 2$ parameters) and $N_e$ columns.



### 2.2.2 Data assimilation scheme

To perform the spatio-temporal data assimilation we employed the deterministic ensemble smoother with multiple data as-
similation (DES-MDA) scheme introduced by Emerick (2018). This is a deterministic version of the original (stochastic)
ensemble smoother with multiple data assimilation (ES-MDA; Emerick and Reynolds, 2013), inspired by the deterministic
ensemble Kalman filter (DEnKF; Sakov and Oke, 2008), which is a deterministic version of the classic (stochastic) ensemble
Kalman filter (EnKF; Evensen, 1994; Burgers et al., 1998). Unlike the stochastic schemes, these deterministic schemes do not
require perturbing the actual or modeled observations (van Leeuwen, 2020), so that sampling errors are reduced while main-
taining sufficient ensemble spread (Sakov and Oke, 2008). The DEnKF thus serves the same purpose as "square root" EnKF
schemes, such as the local ensemble transform Kalman filter (LETKF; Hunt et al., 2007), which are widely used operationally
(Houtekamer and Zhang, 2016), but it is considerably easier to implement. Moreover, it allows for both covariance localization
and subspace inversion (Emerick, 2018). We use a (batch) smoother rather than a filter to allow information from the observa-
tions to propagate backwards in time, since this has been shown to lead to better snowpack reconstruction (Alonso-González
et al., 2022a). In operational snow hydrology settings, notably for forecast initialization, filtering may be preferable due to
computational constraints. The iterative methods used herein could nonetheless readily be extended to filtering (Sakov and
Oke, 2008; Emerick and Reynolds, 2012).





---

**Algorithm 1** DES-MDA with domain localization

---

**Require:** $N_e$ ensemble members, $N_a$ assimilation cycles, $N_g$ grid cells, $N_p$ parameters, $\mathbf{D} = [d_{ij}]$ distance matrix ($N_g \times N_g$), $\mathbf{1}_{N_e}$ vector of ones ($N_e \times 1$), prior parameter ensemble matrix $\mathbf{U}_0$ from Eq. (3)

1: **for** $\ell = 0$ **to** $N_a$ **do**

2:   Run $N_g \times N_e$ forward simulations to obtain the complete spatio-temporal ensemble of internal states $\mathbf{V}_\ell = \mathcal{M}(\mathbf{U}_\ell)$ and predicted observations $\widehat{\mathbf{Y}}_\ell = \mathcal{H}(\mathbf{V}_\ell)$.

3:   **if** $\ell < N_a$ **then**

4:     **for** $i = 1$ **to** $N_g$ **do**

5:       Store the ensemble of parameters for this grid cell $i$ and cycle $\ell$ in the $N_p \times N_e$ matrix $\mathbf{U}_\ell^{(i)}$.

6:       Store the $N_o^{(i)}$ local observations satisfying $d_{ij} < 2c$ in the $N_o^{(i)} \times 1$ vector $\mathbf{y}^{(i)}$

7:       Store the corresponding local predicted observations in the $N_o^{(i)} \times N_e$ matrix $\widehat{\mathbf{Y}}_\ell^{(i)}$

8:       Build the localization matrices $\boldsymbol{\rho}_{\mathbf{U}\widehat{\mathbf{Y}}}^{(i)}$ (size: $N_p \times N_o^{(i)}$) and $\boldsymbol{\rho}_{\widehat{\mathbf{Y}}\widehat{\mathbf{Y}}}^{(i)}$ (size: $N_o^{(i)} \times N_o^{(i)}$) by extracting the corresponding entries from the $N_g \times N_g$ domain-wide GC correlation matrix $\boldsymbol{\rho}$.

9:       Compute the $N_p \times N_o^{(i)}$ local ensemble parameter-predicted observation covariance matrix $\mathbf{C}_{\mathbf{U}\widehat{\mathbf{Y}}}^{(i)} = \frac{1}{N_e}\mathbf{U}^{(i)\prime}\widehat{\mathbf{Y}}^{(i)\prime\mathrm{T}}$ where primes $(\cdot)^\prime$ denote anomalies (deviations from the ensemble mean).

10:      Compute the $N_o^{(i)} \times N_o^{(i)}$ ensemble predicted observation covariance matrix $\mathbf{C}_{\widehat{\mathbf{Y}}\widehat{\mathbf{Y}}}^{(i)} = \frac{1}{N_e}\widehat{\mathbf{Y}}^{(i)\prime}\widehat{\mathbf{Y}}^{(i)\prime\mathrm{T}}$

11:      Compute the $N_p \times N_o^{(i)}$ localized ensemble Kalman gain matrix using subspace inversion (Emerick, 2016)

$$\mathbf{K}_\ell^{(i)} = \left[\boldsymbol{\rho}_{\mathbf{U}\widehat{\mathbf{Y}}}^{(i)} \odot \mathbf{C}_{\mathbf{U}\widehat{\mathbf{Y}}}^{(\ell)}\right]\left(\left[\boldsymbol{\rho}_{\widehat{\mathbf{Y}}\widehat{\mathbf{Y}}}^{(i)} \odot \mathbf{C}_{\widehat{\mathbf{Y}}\widehat{\mathbf{Y}}}^{(\ell)}\right] + \alpha^{(\ell)}\mathbf{R}\right)^{-1}, \tag{4}$$

         where $\mathbf{R} = \sigma_y^2 \mathbf{I}_{N_o^{(i)}}$ is the diagonal local observation error covariance matrix with constant inflation $\alpha^{(\ell)} = N_a$.

12:      Update the ensemble mean $\overline{\mathbf{u}}_\ell^{(i)} = \frac{1}{N_e}\mathbf{U}_\ell^{(i)}\mathbf{1}_{N_e}$ through

$$\overline{\mathbf{u}}_{\ell+1}^{(i)} = \overline{\mathbf{u}}_\ell^{(i)} + \mathbf{K}_\ell^{(i)}\left[\mathbf{y}^{(i)} - \overline{\mathbf{y}}_\ell^{(i)}\right], \tag{5}$$

         where $\overline{\mathbf{y}}_\ell^{(i)} = \frac{1}{N_e}\widehat{\mathbf{Y}}_\ell^{(i)}\mathbf{1}_{N_e}$ contains the ensemble mean predicted observations.

13:      Update the ensemble anomalies $\mathbf{U}_\ell^{(i)\prime} = \mathbf{U}_\ell^{(i)} - \overline{\mathbf{u}}_\ell^{(i)}\mathbf{1}_{N_e}^{\mathrm{T}}$ through

$$\mathbf{U}_{\ell+1}^{(i)\prime} = \mathbf{U}_\ell^{(i)\prime} - 0.5\mathbf{K}_\ell^{(i)}\left[\mathbf{Y}^{(i)} - \widehat{\mathbf{Y}}_\ell^{(i)}\right]. \tag{6}$$

14:      Combine the mean and anomalies to obtain the updated ensemble $\mathbf{U}_{\ell+1}^{(i)} = \overline{\mathbf{u}}_{\ell+1}^{(i)}\mathbf{1}_{N_e}^{\mathrm{T}} + \mathbf{U}_{\ell+1}^{(i)\prime}$

15:    **end for**

16:  **end if**

17: **end for**

---





## 2.3 Experimental setup

In the modeling pipeline, a crucial step is the determination of the distances between grid cells in the simulation domain. This
is typically accomplished through the calculation of the pairwise Euclidean distance between cells. The Euclidean distance
between two cells is the Euclidean norm:

$$d_{ij}^{(E)} = \sqrt{\mathbf{d}_{ij}^{\mathrm{T}} \mathbf{d}_{ij}} \tag{7}$$

where the vector $\mathbf{d}_{ij} = \mathbf{r}_i - \mathbf{r}_j$ is the difference between the generalized coordinates $\mathbf{r} \in \mathbb{R}^K$ of two cells. Using (7) the pairwise
Euclidean distance between all cells in the domain can be computed and stored in a $N_g \times N_g$ symmetric matrix $\mathbf{D} = [d_{ij}]$ with
zeros on the diagonal, which can then be used to define prior correlation (1) and covariance (2) matrices.

As explained in the Introduction, we aim to incorporate topographical dimensions in our distance calculations. Thus, it is
necessary to account for the differences in the units and the potential correlation between these additional dimensions. To do
so, we employ the Mahalanobis distance (e.g. Murphy, 2022) that both standardizes the respective dimensions and takes into
account the correlation between them through a covariance-based normalization. The Mahalanobis distance between two grid
cells is computed as follows

$$d_{ij}^{(M)} = \sqrt{\mathbf{d}_{ij}^{\mathrm{T}} \mathbf{S}^{-1} \mathbf{d}_{ij}} \tag{8}$$

where $\mathbf{S}$ is the $K \times K$ sample covariance matrix computed of the generalized coordinates $\mathbf{r} \in \mathbb{R}^K$ of all the $N_g$ grid cells in
the domain. The use of dimensions other than the usual spatial ones in the definition of $\mathbf{r}$, opens up a vast array of possibilities
that are not just restricted to topographical features. Thus, any characteristic of the domain could be used to map the similarity
between cells, including climatological characteristics such as distance to the ocean as a proxy for continentality at larger
scales and coarser resolutions. Another option is to use remotely sensed observations to define a distance that directly maps the
similar behavior of the cells within a domain in terms of snow-related variables, such as snow cover duration.

In this study, we propose three experiments using different distances to construct the prior covariance and explore the
potential for spatio-temporal snow DA at hyper resolution:

– Experiment I: The prior covariance was constructed using the Euclidean distance in a two horizontal dimensions (easting
and northing) space as is typically done in "3D" land DA (Reichle and Koster, 2003; De Lannoy et al., 2010)

– Experiment II: The prior covariance was constructed using the Mahalanobis distance in a high-dimensional space that
includes three spatial dimensions (easting, northing, and elevation) along with four additional topographic dimensions
described below.

– Experiment III: The prior covariance was constructed using the Mahalanobis distance in a space composed of two
horizontal dimensions and one snow depth dimension based on a snow depth map obtained early in the water year (14
Jan 2020).





In Experiment II, the topographic parameters that define the additional dimensions are the: topographic position index (TPI, with a search distance of 25 m), slope, diurnal anisotropic heating index, and maximum upwind slope index (search distance 15 m, main wind direction $315°$). These topographic parameters (as well as their hyperparameters) capture preferential snow deposition (TPI, wind) and melt (heating index) and were selected based on previous studies in the Izas catchment (Revuelto et al., 2020). Among these experiments, we expect the spatio-temporal snow DA in Experiment III to perform best since it incorporates information from a snow depth map that is directly related to snowpack behaviour to construct the prior covariance. One-time snow depth maps are easier to acquire than annually/monthly repeated surveys and available for more catchments, and Experiment III is thus a plausible setup. In a sense, we are to some extent committing a cardinal sin of circularity in Bayesian inference by using the drone data twice: both for constructing our prior and later for assimilation. Nonetheless, we only actually assimilate a minute fraction $(0.11\%)$ of the data that is used to construct the prior. Moreover, only a single drone-based map enters the prior in Experiment III through a (generalized) distance-based correlation rather than through some form of direct insertion of empirical estimates of prior hyperparameters. Finally, the use of data to construct the prior falls under the domain of empirical Bayes (Efron and Hastie, 2016), which is widely used in both spatial statistics (Cressie and Wikle, 2011) and machine learning (Murphy, 2022). As such, one could classify Experiment III as a (very) mild-form of empirical Bayes.

## 2.4 Computational setup

All the spatio-temporal DA experiments were developed using MuSA. In fact, these new capabilities that we test in these experiments are packaged as an updated version of MuSA where spatio-temporal DA can be activated optionally while preserving the previous capabilities. A variety of different ensemble-based data assimilation schemes were implemented in the original version of MuSA. For this study, we also added the deterministic ES-MDA (DES-MDA Emerick, 2018) to this list. To date, the MuSA system is the first to assimilate drone-based snow depth retrievals (Alonso-González et al., 2022a) and has also been used to study the potential of assimilating land surface temperature to improve snow water equivalent simulations in a synthetic experiment (Alonso-González et al., 2022b).

Several modifications of the MuSA code were necessary to implement spatio-temporal snow DA capabilities. In the original version of MuSA each grid cell in the simulated domain was updated independently. This was due to the fact that both FSM2 and the data assimilation schemes were purely temporal, in the sense that spatial grid cells did not interact, resulting in what is known in computer science as an embarrassingly parallel problem since it makes parallelization relatively trivial. However, spatio-temporal DA requires each grid cell to have access to both the observations and ensemble of predicted observations from nearby cells. What constitutes nearby will depend on the distance metric used (i.e., (7) or (8)) as well as the dimensions of the generalized coordinates $\mathbf{r} \in \mathbb{R}^K$ in which these distances are measured. This posed several computational challenges for the implementation of the spatio-temporal DA. In particular, it required a substantial modification of several MuSA routines that control the timing of the updates of each grid cell to avoid de-synchronizing the spatial simulations, especially in distributed computational facilities.

The MuSA framework was originally designed to be a noninvasive Python wrapper around the supported numerical snowpack model (FSM2) which simplifies implementing updated versions of this model and even altogether new models of any



type. Staying true to this philosophy, the spatial propagation of information was handled using physical disk input/output (I/O) operations. This was designed also with the intention of alleviating the cost of storing many ensemble simulations in memory. Such a cost is possibly prohibitive in applications with a higher spatial density of observations, since each cell would have

to store in memory the ensemble members for all the observed grid cells in its surroundings. Nonetheless, the computational problem became significantly more complex in terms of generating potential bottlenecks due to intensive I/O use. To overcome this problem we improved the performance of several internal routines by decreasing the numerical precision of many variables whenever possible and compressing the binary objects to be shared by I/O operations using the high performance compressor Blosc (Blosc Development Team, 2009-2023). These modifications allow MuSA to run spatial propagation experiments in

both a local (single node) server and on distributed computing clusters where different nodes have to be synchronized, at an affordable cost. However, it is necessary to take into account that the computational cost of the simulation increases considerably when spatio-temporal DA is used.

## 2.5 Evaluation

The results of the three proposed experiments were evaluated using different strategies. A small percentage of the available

grid cells ($0.11\%$ of each map) were included in the assimilation, and therefore the evaluation is essentially performed with independent data despite being compared with the drone data itself. Even so, to ensure completely independent evaluation, the few cells included in the assimilation were not included in the evaluation metrics for the respective experiments.

As a first step, we compared the spatial patterns of snow depth from the simulations with a complete snow depth map retrieved close to peak SWE (11/March/2020). Different metrics were used to estimate the performance of the different ex-

perimental setups. We computed the cell by cell difference (error) between the reference map and the posterior mean of the ensembles. For all non-probabilistic metrics and visualizations, we always used the posterior ensemble mean as the point estimates that we evaluate. To measure the performance of the posterior ensembles, we computed the cell by cell continuous ranked probability score (CRPS Hersbach, 2000) which is a metric for ensemble verification that generalizes the mean absolute error for probabilistic estimates. Due to the large number of grid cells it was not feasible to store the complete spatio-temporal

ensembles. Instead we output the ensemble mean and standard deviation, from which the CRPS was estimated based on a normal approximation of the posterior distribution of the snow depth (Gneiting et al., 2005).

To visually gauge the overall performance of the experiments, we generated scatter plots showing the simulated versus the observed snow depth for all grid cells in the domain. In addition, we computed two commonly used evaluation metrics: the root-mean-square error (RMSE) and the linear correlation coefficient ($r$). In order to check the spatial scale dependence of

the results and how these agree with observations we computed the semivariograms (Chilès and Delfiner, 2012; Wikle et al., 2019) of the posterior mean from the DA experiments, the deterministic open-loop (no assimilation), and the drone map. The semivariogram was computed based on the horizontal Euclidean distance lag (or separation) ranging from 10 m to 150 m using a bin size of 10 m. To evaluate the closeness between the reference semivariogram and the ones simulated by the proposed experiments, we compute the Fréchet distances between them which provide an estimation of the dissimilarity between curves.





Finally, we explored the continuous temporal evolution of the catchment total snow volume in the simulations and compared
it to the evolution of the total snow volume from the 12 drone maps.

## 3    Results

The deterministic OL (open loop) model run was not able to reproduce the intricate spatial patterns of snow depth observed in
the drone surveys (Fig. 3). Almost no spatial variability can be observed, the little variability that exists is likely induced by
the differences in the incoming shortwave radiation simulated by MicroMet. As a result of the poor representation of spatial
variability in the snowpack and the complete lack of snow depths above 2 m, the OL simulation exhibited the worst performance
relative to the observations among all the simulations with a $r = 0.13$ and RMSE= 1.11 m.

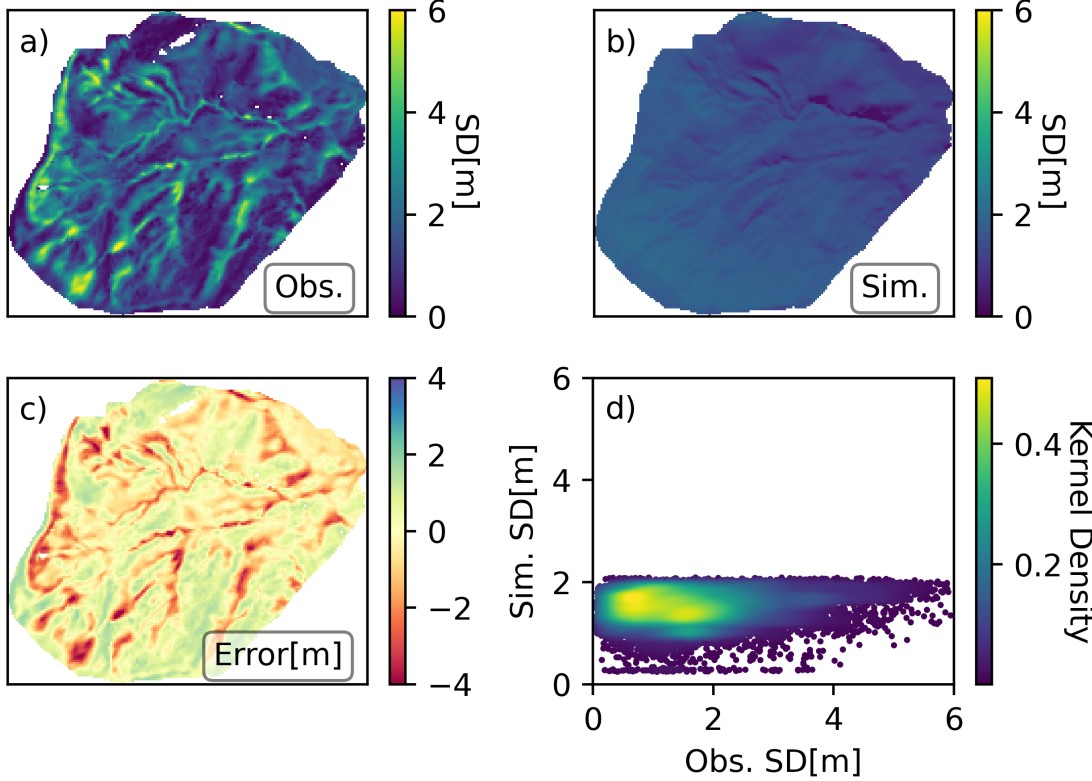

**Figure 3.** a) Observed snow depth map on the 11[th] of March 2020 close to the peak accumulation, b) Deterministic open loop (OL) simulated
snow depth at the same date, c) Difference between the OL and observed map (OL - Obs), and d) Scatter plot showing the OL versus observed
snow depth for each 5 m pixel.



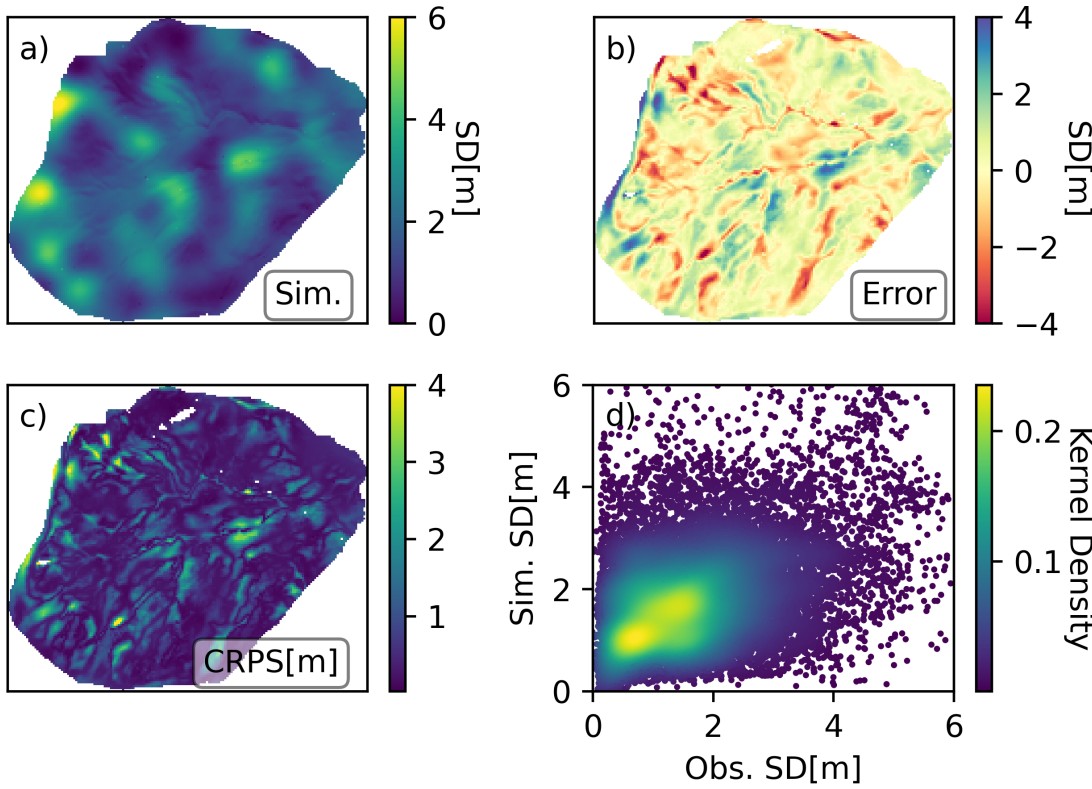

**Figure 4.** Experiment I results on the 11th of March 2020; a) Posterior mean snow depth, b) Difference between the posterior mean and observed snow depth, c) CRPS between the posterior and the observed snow depth , and d) Scatter plot showing the posterior versus observed snow depth for each 5 m pixel.

In all spatio-temporal DA experiments, both the distance-based prior covariance matrix and the localization had a strong impact on the DA performance. The results from Experiment I in Figure 4 show the performance of the spatio-temporal DA when the horizontal Euclidean distance was used to construct the prior covariance and for domain localization. The use of spatio-temporal DA resulted in a posterior snowpack with considerably more spatial variability than the OL. The posterior simulation partly improved the evaluation metrics compared with the OL at least in terms of correlation ($r = 0.38$), albeit not in terms of the error (RMSE $= 1.16$ m). This indicates that although the shapes of the inferred spatial patterns in snow depth are more encouraging than in the OL, the posterior snow depths in Experiment I could still not accurately represent the distribution of the observed snow depth, as shown by the error and CRPS maps (panels b and c). Instead of the intricate spatial patterns of the snow depth observations that reflect wind-redistribution in complex topography, the blob-like Gaussian shape



of the GC correlation function based on horizontal distance is evident in the posterior snow depth maps. As such, the point by point relation between the observations and posterior simulations (panel d) exhibits a large dispersion. This indicates that, at least for hyper-resolution snow simulations, the prior covariance structure was poorly specified in Experiment I.

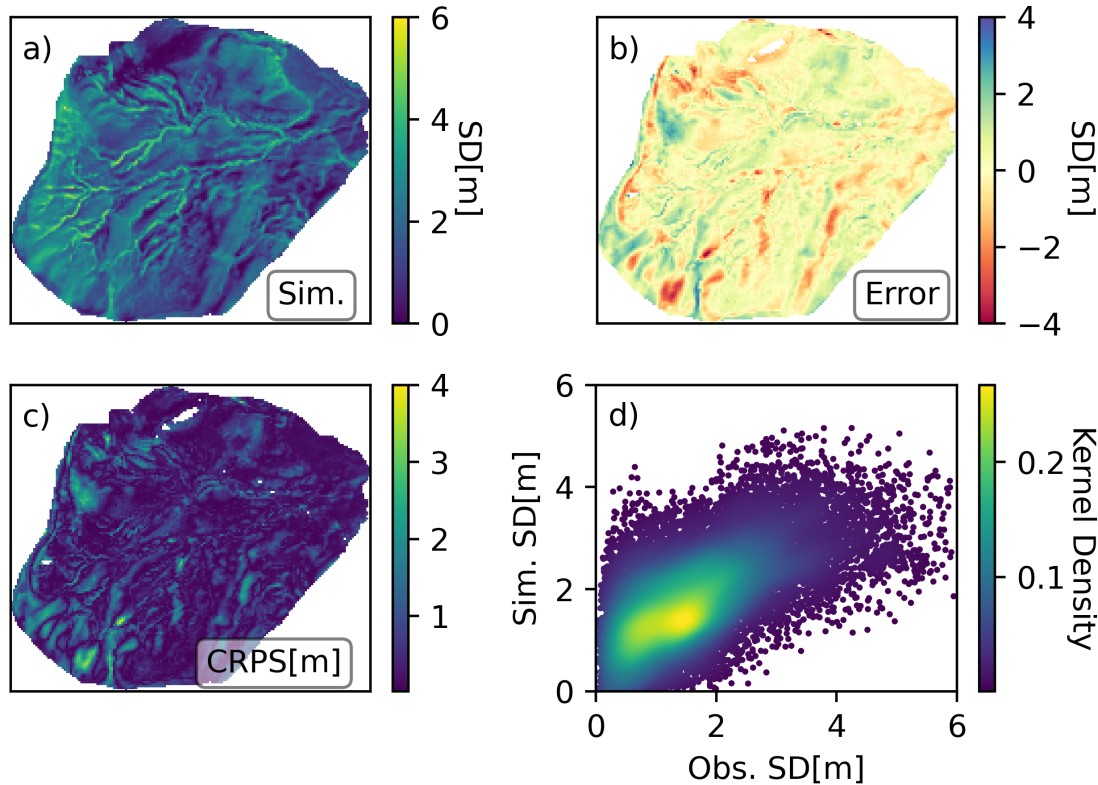

**Figure 5.** Experiment II results on the 11$^{\text{th}}$ of March 2020 presented in the same way as those of Experiment I in Figure 4.

The use of additional dimensions based on topographic parameters in the construction of the prior covariance matrix in Experiment II had a marked positive impact on the performance of the posterior inference (Figure 5). Compared to the OL or the use of horizontal Euclidean distance in Experiment I, the use of topography-based Mahalanobis distance led to a notable improvement in the evaluation and the resulting evaluation metrics with $r = 0.63$ and $\text{RMSE} = 0.89$ m. Furthermore, the incorporation of topographic dimensions enabled the emergence of complex spatial patterns in the simulation, resulting in a 380   more realistic spatial snow depth distribution that more closely aligns with the intricate snow depth patterns shown by the drone-based observations. Despite the improvement in the simulation of spatial patterns in the snow depth distribution, certain obvious limitations are still evident. One notable limitation is the inability to accurately simulate the formation of the cornice at

the north western rim of the catchment. Due to its position at the boundary of the simulation domain, there is a strong domain boundary effect on the maximum upwind slope index, which makes it difficult to simulate this event. The cornice formation
is also influenced by snow drift that is transported from the surrounding area outside of the simulated domain. In general, the discrepancies seem to follow the topographical characteristics of the terrain, as can be seen in the error and CRPS maps.

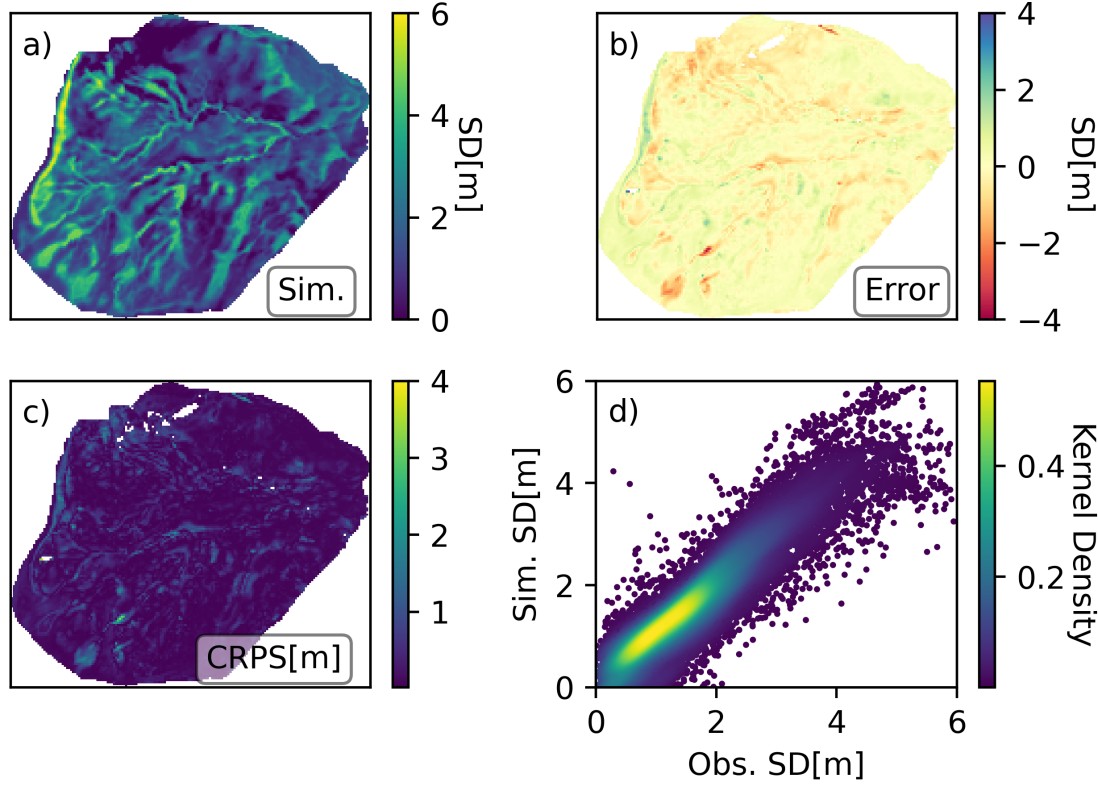

**Figure 6.** Experiment III results on the 11[th] of March 2020 presented in the same way as those of Experiment I in Figure 4.

As expected, the posterior results in Experiment III, where a single drone-based snow depth map (14/Jan/2020) was used to construct the prior covariance, shows by far the most promising results relative to drone observations (11/Mar/2020) as shown in Figure 6. The evaluation metrics were also considerably better than any of the other simulations, with a 46% improvement
in correlation ($r = 0.92$) and a halving of RMSE (RMSE $= 0.44$ m) compared to the second best result in Experiment II. The largest errors shown by the error and CRPS maps seem to be concentrated in very specific locations, exhibiting a mostly homogeneous spatial distribution over the whole area. Experiment III shows that using snow depth observations from early in the snow season to construct a semi-empirical prior covariance matrix to map the similarity between cells can result in spatially





**Table 1.** Summary of validation statistics for each of the snow depth drone maps and the mean for all the season. The highlighted values indicate the best performance value for each date.

| Date | OL | | | | Exp. I | | | | Exp. II | | | | Exp. III | | |
|---|---|---|---|---|---|---|---|---|---|---|---|---|---|---|---|
| | Bias | RMSE | r | CRPS | Bias | RMSE | r | CRPS | Bias | RMSE | r | CRPS | Bias | RMSE | r |
| 14/Jan | -0.06 | 0.69 | 0.18 | 0.41 | 0.09 | 0.73 | 0.33 | 0.32 | 0.13 | 0.55 | 0.65 | **0.11** | **0.05** | **0.22** | **0.95** |
| 03/Feb | -0.06 | 0.81 | 0.17 | 0.48 | 0.17 | 0.87 | 0.36 | 0.38 | 0.18 | 0.65 | 0.65 | **0.09** | **0.05** | **0.20** | **0.97** |
| 24/Feb | -0.38 | 0.87 | 0.19 | 0.46 | -0.13 | 0.81 | 0.39 | 0.35 | **-0.12** | 0.61 | 0.66 | **0.15** | -0.17 | **0.28** | **0.96** |
| 11/Mar | -0.19 | 1.11 | 0.13 | 0.65 | 0.15 | 1.16 | 0.38 | 0.52 | 0.16 | 0.89 | 0.63 | **0.25** | **-0.01** | **0.45** | **0.92** |
| 29/Apr | -0.78 | 1.15 | 0.15 | 0.51 | -0.25 | 0.92 | 0.38 | 0.46 | -0.34 | 0.77 | 0.61 | **0.28** | **-0.24** | **0.46** | **0.90** |
| 03/May | -0.67 | 1.04 | 0.03 | 0.47 | -0.25 | 0.85 | 0.38 | 0.50 | -0.34 | 0.71 | 0.61 | **0.34** | **-0.23** | **0.43** | **0.89** |
| 12/May | -0.17 | 0.50 | 0.00 | 0.31 | 0.15 | 0.69 | 0.27 | **0.30** | **0.07** | **0.50** | 0.52 | 0.39 | 0.19 | 0.54 | **0.75** |
| 19/May | -0.30 | 0.57 | 0.00 | **0.31** | -0.03 | 0.62 | 0.35 | 0.35 | -0.13 | 0.48 | 0.51 | 0.39 | **0.01** | 0.42 | **0.80** |
| 26/May | **0.00** | **0.31** | 0.18 | 0.18 | 0.15 | 0.47 | 0.29 | **0.15** | 0.08 | 0.32 | 0.38 | 0.16 | 0.16 | 0.37 | **0.66** |
| 02/Jun | -0.02 | **0.18** | 0.14 | 0.09 | 0.06 | 0.33 | 0.23 | **0.06** | **0.00** | 0.20 | 0.25 | 0.07 | 0.04 | 0.23 | **0.52** |
| 10/Jun | **0.01** | **0.11** | 0.00 | 0.16 | 0.07 | 0.28 | 0.14 | **0.13** | 0.03 | 0.13 | 0.15 | 0.23 | 0.04 | 0.17 | **0.31** |
| 21/Jun | 0.01 | 0.07 | 0.00 | 0.14 | 0.05 | 0.21 | 0.05 | **0.10** | **0.01** | **0.08** | 0.03 | 0.14 | 0.02 | 0.09 | **0.14** |
| mean | -0.22 | 0.62 | 0.09 | 0.35 | 0.02 | 0.66 | 0.30 | 0.30 | -0.02 | 0.49 | 0.47 | **0.22** | **-0.01** | **0.32** | **0.73** |

complex posterior simulations that closely align with the peak snow depth spatial patterns found in the independent validation
data. This is reflected in all evaluation metrics, which are markedly improved using this approach compared not only to the OL
but also to the alternative spatio-temporal DA strategies tested in Experiments I & II. Table 1 provides a detailed description
of the spatial validation metrics for maps other than the peak SWE occurrence date. All experiments show decreasing $r$ values
over time. This is due to the very shallow snowpacks found at the end of the season such that small absolute differences in the
snow depth are exaggerated. The patchy conditions at the end of the season also contribute to this decrease in performance. In
any case, all experiments were able to greatly improve the statistics during the melt season compared to the OL, where melting
occurred much earlier, particularly Experiment III which maintained $r > 0.5$ until early June. After that date only reminiscent
accumulations were left.

Figure 6 shows the spatial variability in snow depth at different spatial scales estimated for the three experiments, the
OL, and the observations close to peak SWE (11/Mar/2020). As mentioned before, the OL simulation exhibits very limited
spatial variability compared with the observations, and this is reflected in the semivariogram which shows is nearly flat and
thus shows minimal dependence on separation distance. On the other hand, the three experiments show increasingly realistic
semivariograms depending on the level of spatio-temporal DA complexity. The shape of the semivarigorams nonetheless differs
considerably between the three experiments. Among the three experiments, the semivariogram for Experiment I semivariogram
is the furthest from the reference (observed) semivariogram according to the Fréchet distance metric (FD = 0.71), despite
showing a noticeable improvement over the OL (FD = 1.07). In particular, Experiment I semivariogram is constrained to the



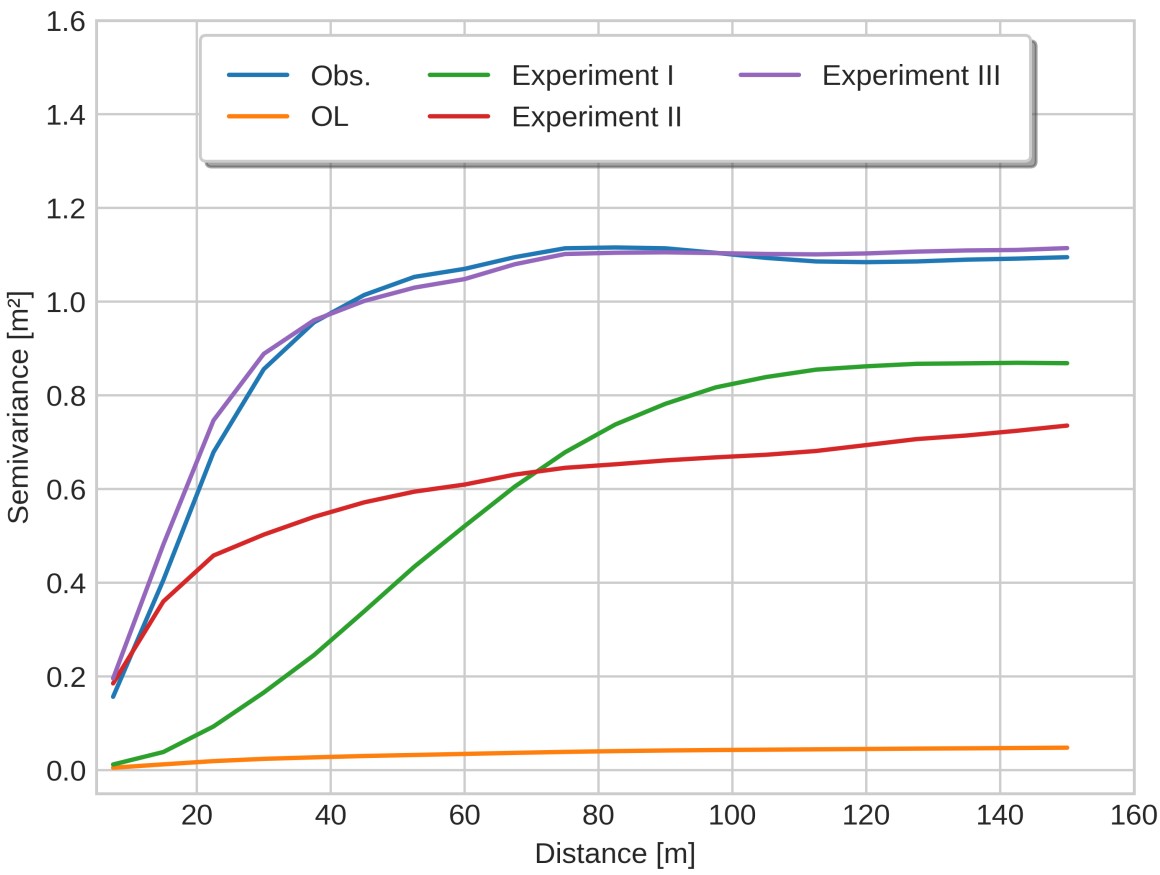

**Figure 7.** Comparison of snow depth semivariograms on the 11[th] of March 2020 near peak accumulation.

Gaussian shape characteristic of the GC function, as this is implemented directly in the hoirzontal Euclidean distance which is directly related to the $x$-axis in Figure 7. In geostatistical terms (c.f. Chilès and Delfiner, 2012; Wikle et al., 2019), the semivariogram for Experiment I diverges from the observations in two important respects. Firstly, it does not show the correct high small-scale variability since it lacks the so-called nugget effect (offset around 0 distance). Secondly, despite performing better than the OL, it displays lower large scale variability than the observations in that it has a lower sill (stabilizes at a lower value) than the observed semivariogram. In Experiment II the semivariogram is able to capture the same nugget effect as in the observations, but the larger scale variability remains too low with a sill that is similar to that that in Experiment I. Overall, the semivariogram is nonetheless closer to the observations in Experiment II (FD= 0.47) than in Experiment I. As could be expected, the use of observations from an earlier drone-based snow depth map to define the prior covariance between cells



in Experiment III also led to the most realistic scale-dependence in the simulation spatial patterns of snow depth according
to the semivariogram. Here both the nugget effect, the sill, as well as the overall shape of the semivariogram were more or
less in complete agreement. Notably, the semivariogram of Experiment III showed the shortest Fréchet distance (FD = 0.08),
confirming a very close match to the observed steep increase in the semivariance of snow depth as a function of separation
distance near peak accumulation in 2020.

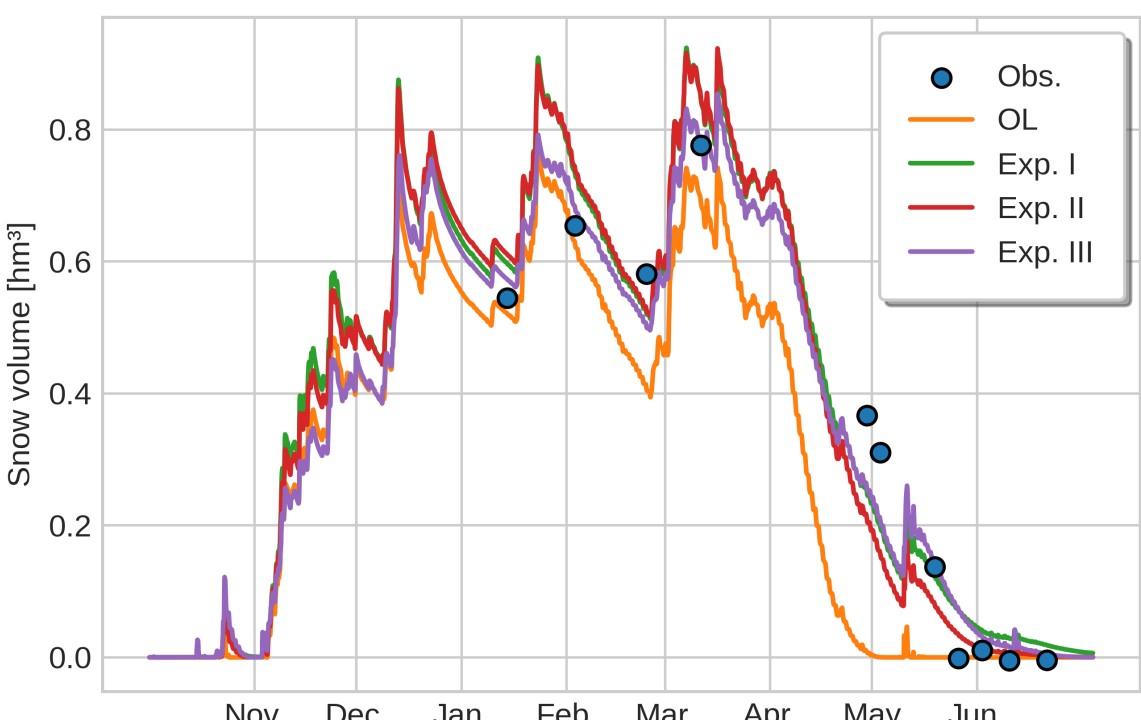

**Figure 8.** Time evolution of the total volume of snow in the catchment from the different simulations (colored lines) along with the volume
estimates obtained from the snow depth drone maps (blue dots).

The improvements in the simulated spatial patterns that we saw near the peak snow depth date for the various experiments
is also reflected in the temporal evolution of the total snow volume in the catchment. Figure 8 compares the temporal evolu-
tion of the total volume of snow in the Izas catchment as simulated continuously by in the OL and the 3 spatio-temporal DA
experiments with the snapshots retrieved from the drone surveys. Due to the increasing variability induced by the DA, some
parts of the basin show large accumulations, which delays the melt out date and makes it more consistent with the trajectory



of the effectively independent observations. All the experiments improved the (temporal) RMSE metric for the posterior mean total snow volume obtaining RMSE $= 0.07$, $0.08$ and $0.06$ hm$^3$ for the posterior mean in experiments I, II and III, respectively, compared with a considerably higher RMSE $= 0.16$ hm$^3$ for the OL. This shows how spatio-temporal DA of sparse observations can lead to a better representation of both total snow accumulation and snowmelt timing at the catchment level, which could in turn improve snowmelt runoff simulations. Surprisingly, despite the improvements shown by the Experiments II & III

compared with the Experiment I in terms of spatial representativity, the temporal RMSE values of in the three experiments did not differ markedly. As such, even a relatively simple spatio-temporal DA system based on horizontal distance in experiment I may suffice if the primary quantity of interest is the temporal evolution of the total snow volume (or possibly by extension SWE) in a catchment, rather than accurately reproducing hte spatial distribution of snow depth.

## 4    Discussion

In this work, we investigated the capability of propagating information from sparse observations of the snowpack in hyper-resolution simulations through ensemble-based spatio-temporal data assimilation techniques. We performed three experiments in which we assimilated sparse observations from the Izas experimental catchment. The observations were obtained through random sampling of 20 points from each of the 12 available snow depth maps during the 2020 water year. This setup was designed to mimic the typical sparse manual sampling of a catchment to test the possibility of propagating this information

in distributed snowpack simulations. It should be noted that the selection of grid points for sampling is completely random, and different from one date to another, which may result in many measurements not being as informative as they could be. Additionally, most of the snow depth maps are concentrated on the end of snow season, at which point a significant portion of the snow has already melted (Fig. 1), leading to a partial sampling of snow free areas. While the absence of snow at a specific date can provide valuable information in the context of snow DA (e.g. Margulis et al., 2016; Fiddes et al., 2019; Alonso-

González et al., 2021), in the absence of clouds this information is already available from fSCA retrieved by high resolution optical satellite imagery (Aalstad et al., 2020). In practice it could thus be seen as somewhat of a waste to choose to sample snow free areas with manual measurements when we could instead assimilate fSCA from satellites. The joint assimilation of fSCA and sparse snow depth observations is a certainly promising path forward, that will be explored in future work. In real-world scenarios, it would also be beneficial to implement more sophisticated sampling strategies to maximize the expected

information gain (Molotch and Bales, 2005; López-Moreno et al., 2011a), but for the sake of simplicity the current completely random approach is deemed adequate for our intended objectives.

The three experiments that we carried out herein reflect different strengths and weaknesses of spatio-temporal DA techniques when applied to the hyper-resolution scales for snowpack simulation. These three experiments were designed as a sample of the potential of the techniques, rather than to find the optimal setup which would likely be highly problem-dependent and

involve considerable hyperparameter tuning. The configuration used in Experiment I was not able to reproduce the complex spatial patterns present in the drone-based snow depth map near the snow accumulation maximum, although the simulated evolution of the total snow volume was similar to that of the other two DA experiments and equally close to the observations.





Despite this weakness, the simplicity of the configuration of Experiment I is a key advantage over the more sophisticated prior covariance modeling experiments. Constructing the prior covariance based solely on the (horizontal) geographic distance between cells allows for a much more intuitive configuration of the correlation length scale $c$ in the GC function (1). Tuning such hyperparameters by trial and error can be prohibitively computationally expensive depending on the size of the experiment being carried out (Anderson, 2012). Furthermore, the use of distances defined in geographic space can be particularly valuable in the more ideal case of a very high spatial density of observations, where a fine control of spatial information propagation may be desired. Another clear use case of regular geographic-distance based prior covariance modeling and localization can be in the case of lower spatial resolution simulations ($\geq 1$ km), which may be required for larger scale snow data assimilation applications (e.g. Magnusson et al., 2014; Odry et al., 2022), as is typical in land (Reichle and Koster, 2003; De Lannoy et al., 2010, 2012) and atmospheric DA (Anderson, 2012; Houtekamer and Zhang, 2016).

The incorporation of other dimensions than the geographical easting and northing in the distance-based prior covariance and localization markedly improves the results of Experiments II and III both compared to the OL and Experiment I. Experiment II has demonstrated (see Figure 5) that by simply utilizing parameters derived from topography, it is possible to enhance the hyper-resolution simulations such that they can partly reproduce the complex spatial patterns of the snowpack. This is supported by the reasonable assumption that the snowpack often exhibits persistent spatial patterns between seasons (Schirmer et al., 2011) due to the topographic controls on snowpack redistribution (Revuelto et al., 2014). Other promising experimental configurations could include additional topography-based parameters, such as avalanche deposition zones (López-Moreno et al., 2017) or sky view factor (Sicart et al., 2006) as they are both important contributors to the mass and energy balances. For simulations at a coarser resolution, we hypothesize that the use of other parameters with climatological implications could play a crucial role. These could be indicators of continentality or marine influences, as well as exposure to atmospheric moisture advection (Alonso-González et al., 2021) or teleconnection indices (López-Moreno et al., 2011b), which have been shown to play an important role in controlling large scale snowpack dynamics. In summary, the approach in Experiment II paves the way towards integrating additional information, such as that derived from (snow-off) DEMs, that can otherwise be difficult to exploit in the context of snow DA.

The results obtained in Experiment III (see Figure 6) were by a good measure the closest to the reference observations used for evaluation. The empirical Bayes approach (Efron and Hastie, 2016) of using observations in the construction of the prior covariance is thus a highly effective way to directly map the similarity between cells. The case shown in Experiment III is likely an example where the benefits from this technique are maximized, as distances are defined based on a complete drone-based snow depth map, a snowpack state variable that is closely related with SWE. It should be noted that nowadays, it is possible to obtain hyper-resolution snow depth maps also for larger or more remote catchments from satellite products such as from the Pléiades constellation (Marti et al., 2016) that can be scheduled in advance. Given the high inter-annual similarity of the snowpack spatial distribution (Schirmer et al., 2011), the snow depth map used for the prior covariance could be from a different year and possibly still perform nearly as well as in our Experiment III. Alternatively, other observation-based products such as satellite-based snow cover duration maps could serve as a proxy for SWE patterns and thus help with prior covariance modeling both at larger scales and at coarser resolutions. Prior covariance modeling can also provide a way to integrate other



sources of information into the snow DA. An example may be the Landsat constellation where the low revisit frequency may limit its direct usage in a snow DA context (Alonso-González et al., 2022b), but the long climate data record can be used to
compute long-term snow climate statistics that could be used to map the similarity between grid cells (Macander et al., 2015).

The spatio-temporal DA techniques that we explored herein also have wider implications. An immediate possible operational application could be to integrate information obtained from the typically sparse national snow monitoring networks into high resolution distributed physically-based snow simulations, building on the work of Magnusson et al. (2014); Cluzet et al. (2022), as an alternative to approaches based purely on statistical interpolation (Fassnacht et al., 2003; Collados-Lara et al., 2020). In
a similar vein, as an extension of the work of Odry et al. (2022), the spatio-temporal snow DA approach presented here allows for the fusion of sparse manual snow surveys into high-resolution distributed snowpack simulations. In a broader sense, these techniques could be used to propagate information obtained from optical satellites that often contain spatial gaps for several reasons. For example, as an extension to the study of De Lannoy et al. (2012), it would be possible to propagate information from clear observed cells to those that lack observations due to the presence of clouds. Propagating information
from forest clearings to areas where canopy obstructs snow visibility from space could also be a promising approach that should be investigated in future studies. More generally, the techniques can be employed to propagate information from areas where information is easily obtained from optical satellites to areas where obtaining information is problematic due to terrain characteristics such as steep slopes or the presence of shadows (Gascoin et al., 2019). The use of spatio-temporal DA techniques in these contexts would allow for controlling the influence of outliers in the observations, increasing the spatial consistency of
the posterior simulations. The use of DA techniques with localization (Sakov and Oke, 2008) also impedes distant observations from affecting local simulations thus avoiding spurious correlations that can degrade the quality of the analysis. The ensemble Kalman-based DA approach pursued herein could also be adopted as a proposal distribution in particle-based DA (e.g. Pirk et al., 2022), such that our approach may also be relevant to particle-based snow DA frameworks (Cluzet et al., 2021, e.g.).

A particularly promising potential application is the assimilation of snow depth acquisitions from the ICESat-2 laser altimeter
(Enderlin et al., 2022; Deschamps-Berger et al., 2022), which records data along linear tracks that exhibit a discontinuous pattern in space. A straightforward extension of the experiments herein could involve joint data assimilation using (nearly) spatially continuous satellite retrievals together with sparser retrievals or ground-based measurements. This could for instance involve combining high spatial resolution fSCA products (Gascoin et al., 2020; Aalstad et al., 2020) derived from Sentinel-2 and/or Landsat acquisitions with in-situ information obtained from stations or manual surveys, or even ICESat-2 snow depth
retrievals. This would exploit the information derived from complementary sources within the same simulation, as done for glacier data assimilation in Leclercq et al. (2017).

Despite the promising results, new configuration and numerical challenges arise when a high-dimensional space is used to define the pairwise distance between cells. For example, the Mahalanobis distance (8) is dimensionless, which impedes an intuitive interpretation of the prior correlation length scale $c$. This, along with the aforementioned computational cost associated with tuning the hyperparameter $c$ by trial and error, further hinders an optimal and automated configuration of the
entire assimilation workflow. A possible solution to avoid the computational cost of adjusting the $c$ hyperparameter could be to explore the semivariogram of the available observations and deduce its value from it (for either the Euclidean or Mahalanobis





distance). However, there is another issue that may complicate the use of the Mahalanobis distance: if simple geographic Euclidean distance is not used, it is not guaranteed that the resulting correlation matrix will be numerically positive definite (Curriero, 2006). The Cholesky decomposition used in (3) is only possible if the prior covariance matrix is positive definite, which can thus make some configurations of dimensions in combination with some values of $c$ not possible. Fortunately, this is not a new problem within the geostatistics (Chilès and Delfiner, 2012), spatio-temporal statistics (Cressie and Wikle, 2011; Wikle et al., 2019), or machine learning (Rasmussen and Williams, 2005; Murphy, 2022, 2023) communities. Different procedures have been proposed to address this issue, including the use of dimension reductions through multi-dimensional scaling (Murphy, 2022). Other approaches such as finding the closest positive definite matrix have been proposed (Davis and Curriero, 2019).The use of specific prior correlation models other than the GC and different approaches to robustly ensure the positive definiteness of the covariance matrix will be explored in future work.

## 5 Conclusions

In this study, we have explored the potential of spatio-temporal DA methods for updating hyper-resolution simulations of the snowpack in an experimental catchment situated in the Pyrenees. Three different experiments were proposed and executed, each employing a distinct prior covariance modeling strategy to assimilating sparse snowpack observations that were subsampled from drone-based snow depth maps. The assimilated data consists of 20 randomly selected snow depth measurements obtained from the $N_g = 18\,442$ possible grid cells (excluding some minor gaps) that were mapped for each of the 12 drone flights. This sampling strategy aims to emulate several manual snow surveys performed intermittently during a water year.

The three proposed experiments are essentially built on the same underlying DA scheme (Algorithm 1), the only difference being in the distance calculation used for the prior covariance matrix and localization. Experiment I utilized the conventional horizontal Euclidean distance between cells following the usual setup of 3D land data assimilation. The results of this first experiment demonstrated an inability to accurately simulate the complex spatial snow depth patterns found in the drone-based observations. Subsequently, Experiments II and III significantly enhanced the performance of the DA system compared to Experiment I, by incorporating additional dimensions in the form of topographic parameters and a snow depth map, respectively, in the distances used to construct the prior covariance. Experiment III performed best by most accurately reproducing the snow depth spatial distribution. At the catchment scale, all three experimental setups led to nearly equally and relatively accurate simulations of the temporal evolution of total snow volume, with a considerable improvement over the deterministic open loop (without data assimilation) both in terms of peak accumulation and snowmelt timing.

It should be noted that setting up the better performing Experiments (II & III) can entail considerable technical difficulties. In particular, the use of generalized (rather than simpler geographic) multi-dimensional distances to construct the prior covariance matrix and perform localization leads to a less intuitive experimental design. Notably, the performance of these experiments is sensitive to the choice of hyperparameters, especially the correlation length scale $c$. These parameters can be difficult to get a sense for when working with generalized distances. Moreover, the computational cost of performing hyperparameter optimization can be prohibitive and some choices for the hyperparameters can even result in non-trivial numerical issues





related to matrix positive definiteness. Further research is necessary to develop and refine the covariance functions utilized in data assimilation to ensure compatibility with hyper-resolution scenarios that require non-conventional distance metrics to exploit and propagate information from observations. Despite the new challenges encountered, the results are promising and pave the way for improving hyperresolution snow simulations through ensemble based assimilation of spatially sparse

data. The methods presented here have many new applications, which will allow the combination of spatially incomplete but potentially very informative data such as that obtained from automatic gound based monitoring networks or orbital LiDAR, with continuous satellite products such as FSCA or LST within the same hyper-resolution distributed simulations.

*Code and data availability.* The MuSA code can be found at https://github.com/ealonsogzl/MuSA (last access: 8 May 2023, Alonso-González (2023)). The complete drone surveys and meteorological forcing can be downloaded from https://doi.org/10.5281/zenodo.7248635 (last ac-

cess: 8 May 2023, Alonso-González (2022)).

## Appendix A: Local information propagation using prior dependence

Herein, we outline the role that prior dependence plays in propagating information from local observations. In A1 we show formally how prior dependence is the only way to propagate information from observed to unobserved locations in the general Bayesian data assimilation setting. In A2 we go through an illustrative toy example in the form of a simple Gaussian model

that demonstrates how observed information can be propagated from an observed to an unobserved location using prior dependence. Distance-based prior dependence modeling is at the core of Gaussian process regression (Rasmussen and Williams, 2005) which Kalman methods can be seen as a special case of. Thereby, the simple Gaussian linear model is an elementary demonstration of the inferential mechanism that powers these Gaussian spatiotemporal techniques (Cressie and Wikle, 2011; Wikle et al., 2019). Many applied spatio-temporal problems involving Gaussian processes, including the one in this study, are

of course non-linear. The simple Gaussian linear model can nonetheless be helpful to gain a high level intuition of how the inference works.

### A1    General Bayesian formulation

DA can be formalized as an exercise in Bayesian inference (Wikle and Berliner, 2007; Evensen et al., 2022) that involves updating prior beliefs $p(\mathbf{x})$ about model states (and/or parameters) using information from observations encoded in the likelihood

$p(\mathbf{y}|\mathbf{x})$ to obtain the posterior

$$p(\mathbf{x}|\mathbf{y}) = p(\mathbf{y}|\mathbf{x})p(\mathbf{x})/Z \,, \tag{A1}$$

where the model evidence $Z = p(\mathbf{y})$ is a normalizing constant (MacKay, 2003). Now split the state space in two regions $\mathbf{x} = [\mathbf{x}_1, \mathbf{x}_2]$ where the first ($\mathbf{x}_1$) is directly observed via local observations $\mathbf{y}_1$ while the second ($\mathbf{x}_2$) is unobserved. In general, these two regions could have different spatio-temporal extents. The likelihood becomes $p(\mathbf{y}|\mathbf{x}) = p(\mathbf{y}_1|\mathbf{x}_1)$ so the joint posterior



(A1) is

$$p(\mathbf{x}_1, \mathbf{x}_2 | \mathbf{y}_1) = p(\mathbf{y}_1 | \mathbf{x}_1) p(\mathbf{x}_1) p(\mathbf{x}_2 | \mathbf{x}_1) / Z, \tag{A2}$$

where we have factorized the joint prior $p(\mathbf{x}) = p(\mathbf{x}_1, \mathbf{x}_2) = p(\mathbf{x}_2 | \mathbf{x}_1) p(\mathbf{x}_1)$. Since the marginal posterior for $\mathbf{x}_1$ is

$$p(\mathbf{x}_1 | \mathbf{y}_1) = p(\mathbf{y}_1 | \mathbf{x}_1) p(\mathbf{x}_1) / Z, \tag{A3}$$

then (A2) can be written more compactly as $p(\mathbf{x}_1, \mathbf{x}_2 | \mathbf{y}_1) = p(\mathbf{x}_1 | \mathbf{y}_1) p(\mathbf{x}_2 | \mathbf{x}_1)$ such that the marginal posterior for $\mathbf{x}_2$ is

$$p(\mathbf{x}_2 | \mathbf{y}_1) = \int p(\mathbf{x}_1, \mathbf{x}_2 | \mathbf{y}_1) \, \mathrm{d}\mathbf{x}_1 = \int p(\mathbf{x}_1 | \mathbf{y}_1) p(\mathbf{x}_2 | \mathbf{x}_1) \, \mathrm{d}\mathbf{x}_1, \tag{A4}$$

demonstrating that local information from $\mathbf{y}_1$ is transferred to $\mathbf{x}_2$ through the prior dependence encoded in $p(\mathbf{x}_2 | \mathbf{x}_1)$. Only in the *special case* of prior independence, i.e. $p(\mathbf{x}_2 | \mathbf{x}_1) = p(\mathbf{x}_2)$, do we have that

$$p(\mathbf{x}_2 | \mathbf{y}_1) = p(\mathbf{x}_2) \int p(\mathbf{x}_1 | \mathbf{y}_1) \, \mathrm{d}\mathbf{x}_1 = p(\mathbf{x}_2), \tag{A5}$$

whereby observing $\mathbf{y}_1$ does not provide any information about $\mathbf{x}_2$, highlighting the vital role that prior dependence plays in propagating information from local observations in general Bayesian DA. This will apply to any scheme that attempts to implement such a propagation in practice, whether it be an ensemble Kalman, particle, MCMC, variational, or hybrid method.

### A2 Simple Gaussian linear example

To make the above general derivation more concrete, we consider a specific toy example in the form of a simple model where a scalar variable of interest $x_i$, such as snow depth or air temperature, at location $i = 1$ is directly observed with some observation error while the same variable is unobserved at a second location $i = 2$. The task now is to infer the value of $x_2$ using noisy observations of $x_1$. Using the following forward (data generating) model for the noisy observation $y_1 = x_1^\star + \varepsilon$ where $x_1^\star$ is the (unknown) true value at the observed location and $\varepsilon \sim \mathcal{N}(0, \sigma_y^2)$ is a zero-mean additive Gaussian observation error with observation error variance $\sigma_y^2$ leads to a Gaussian likelihood $p(y_1 | x_1) = \mathcal{N}(y_1 | x_1, \sigma_y^2)$ of the form

$$p(y_1 | x_1) \propto \exp\left( -\frac{1}{2} \sigma_y^{-2} \left( y_1 - x_1 \right)^2 \right) \tag{A6}$$

which, given this particular forward model, does not depend on $x_2$. We also adopt a bivariate Gaussian joint prior distribution to encode our uncertainty about the values of $x_1$ and $x_2$ of the form

$$p(x_1, x_2) \propto \exp\left( -\frac{1}{2} \left[\mathbf{x} - \boldsymbol{\mu}\right]^{\mathrm{T}} \mathbf{C}^{-1} \left[\mathbf{x} - \boldsymbol{\mu}\right] \right) \tag{A7}$$

where $\mathbf{x} - \boldsymbol{\mu} = \begin{bmatrix} x_1 - \mu_1 \\ x_2 - \mu_2 \end{bmatrix} = \begin{bmatrix} x_1' \\ x_2' \end{bmatrix}$ are the anomalies from the prior means $(\mu_1, \mu_2)$ and $\mathbf{C} = \begin{bmatrix} \sigma_x^2 & \rho\sigma_x^2 \\ \rho\sigma_x^2 & \sigma_x^2 \end{bmatrix}$ is the prior co-variance matrix with variance $\sigma_x^2$ and correlation $\rho$ whose inverse is the prior precision matrix $\mathbf{C}^{-1} = \mathrm{adj}(\mathbf{C})/\mathrm{det}(\mathbf{C}) =$





$\begin{bmatrix} \phi & -\rho\phi \\ -\rho\phi & \phi \end{bmatrix}$ where $\phi = \left[(1-\rho^2)\sigma_x^2\right]^{-1}$. Expanding the exponent in the prior (A7) and multiplying the prior with the likelihood (A6), the posterior becomes

$$p(x_1,x_2|y_1) \propto p(y_1|x_1)p(x_1,x_2) \propto \exp\left(-0.5\sigma_y^{-2}(y_1-x_1)^2 - 0.5\phi\left[\phi x_1'^2 + \phi x_2'^2\right] + \rho\phi x_1' x_2'\right) \qquad (A8)$$

Since the model is linear with a Gaussian prior and Gaussian observation error, the posterior is also Gaussian. Thereby, the mean will coincide with the mode which happens to be the unique point at which the gradient of a Gaussian vanishes. One

simple way to obtain the posterior mean $\mathbf{m} = \begin{bmatrix} m_1, & m_2 \end{bmatrix}^{\mathrm{T}}$ analytically is thus to compute the gradient of the posterior $\frac{\partial p}{\partial \mathbf{x}}$ and identify the mean as the point at which it is zero. Taking this route, the posterior mean at the unobserved location is

$$m_2 = \mu_2 + \rho(m_1 - \mu_1), \qquad (A9)$$

and the posterior mean at the observed location is

$$m_1 = \mu_1 + K(y_1 - \mu_1), \qquad (A10)$$

where $K = \sigma_x^2\left(\sigma_x^2 + \sigma_y^2\right)^{-1}$ is a Kalman gain $K \in (0,1)$ which is determined by the ratio of the prior variance $\sigma_x^2$ to the observation error variance $\sigma_y^2$. Using second derivatives to compute the Hessian (MacKay, 2003), we also obtain the posterior precision matrix $\mathbf{P}^{-1} = \begin{bmatrix} \sigma_y^{-2} + \phi & -\rho\phi \\ -\rho\phi & \phi \end{bmatrix}$ whose inverse is the posterior covariance matrix

$$\mathbf{P} = \begin{bmatrix} K\sigma_y^2 & K\rho\sigma_y^2 \\ K\rho\sigma_y^2 & K\left[\left(1-\rho^2\right)\sigma_x^2 + \sigma_y^2\right] \end{bmatrix} \qquad (A11)$$

whereby the exact joint posterior is given by the bivariate Gaussian $p(\mathbf{x}|y_1) = \mathcal{N}(\mathbf{x}|\mathbf{m}, \mathbf{P})$ with marginal distributions $p(x_1|y_1) =$

$\mathcal{N}(x_1|m_1, K\sigma_y^2)$ and $p(x_2|y_1) = \mathcal{N}(x_2|m_2, K\left[\left(1-\rho^2\right)\sigma_x^2 + \sigma_y^2\right])$ where the marginal variances are given by the corresponding diagonal terms in $\mathbf{P}$. Similar derivations for more general Gaussian linear systems can be found in Murphy (2023).

The point of deriving the exact posterior for this simple toy bivariate model is to demonstrate the importance of prior dependence, which in the case of the Gaussian prior is controlled through the prior correlation $\rho$, in propagating information. First consider the special case of prior independence which occurs when $\rho = 0$. In this case, the posterior mean at the unobserved

location $m_2$ in (A9) reverts to the prior mean $\mu_2$. Moreover, the posterior variance at the unobserved location given by the final diagonal term in (A11) becomes $K[\sigma_x^2 + \sigma_y^2] = \sigma_x^2$ which is simply the prior variance. As such, the observations at the observed location have no effect on the posterior inference at the unobserved location. At the observed location, however, the observation has an effect on the inference with the posterior mean $m_1$ in (A10) pulling from the prior mean $\mu_1$ towards $y_1$ in accordance with the Kalman gain. Similarly, the posterior variance at the observed location given by the initial diagonal term

in (A11) is $K\sigma_y^2$ which will always be less than the prior variance $\sigma_x^2$. As such, unsurprisingly, the observations constrain the posterior at the observed location. This case is shown in Figure A1 where we see that the marginal posterior at the observed




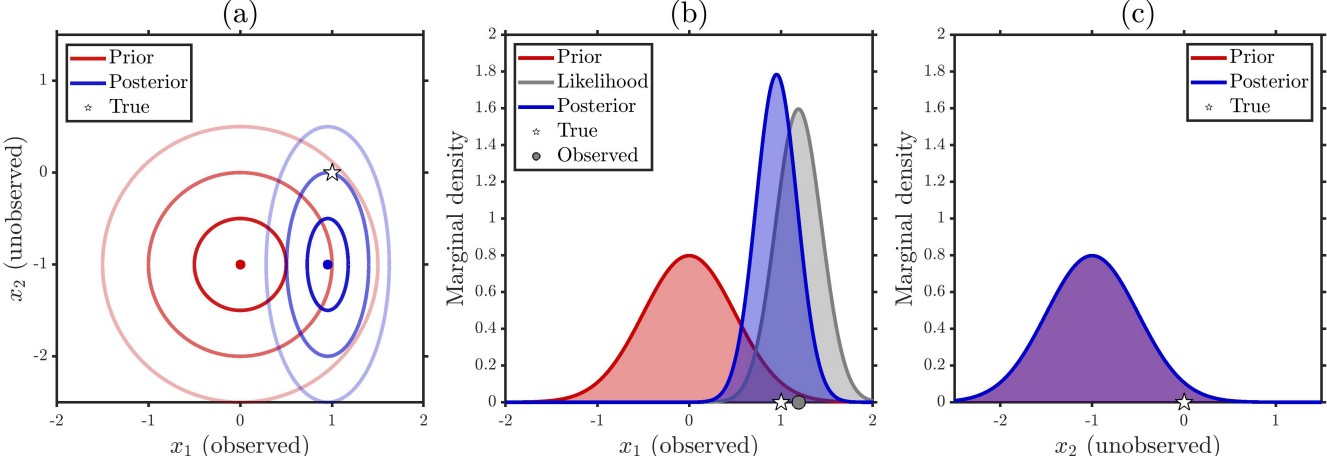

**Figure A1.** Simple Gaussian linear example with an uncorrelated prior $\rho = 0$, prior means $\mu_1 = 0$ and $\mu_2 = -1$, equal prior variances $\sigma_x^2 = 0.25$, observation error variance $\sigma_y^2 = 0.0625$, and observation $y_1 = 1.1911$. Panel (a): Joint prior (red) and posterior (blue) means (dots) and $68\%$ ($1\sigma$), $95\%$ ($2\sigma$), and $99.7\%$ ($3\sigma$) highest density credible intervals (innermost to outermost ellipses), and true value (star). Panel (b): Marginal prior (red curve), marginal posterior (blue), (scaled) likelihood (gray), true value (star), and observation (gray dot) for the observed location. Panel (c): Same as (b) but for the unobserved location.

location is both narrower and closer to the observations than the prior, whereas at the unobserved location the marginal prior and posterior are identical.

For the more general case of prior dependence obtained when $\rho \neq 0$ we see information propagate from the observed to
the unobserved location. As shown in (A9) the update (i.e., $m_2 - \mu_2$) in the posterior mean at the unobserved location is proportional to the update in the posterior mean at the observed location with a constant of proportionality equal to the prior correlation $\rho$. Moreover, the posterior variance at the unobserved location $K[(1-\rho^2)\sigma_x^2 + \sigma_y^2]$ will always be less than the prior variance $\sigma_x^2$ (since $\rho^2 > 0$). Thus, it is clear that with prior correlation the marginal posterior in the unobserved location will be both shifted and constrained (narrowed). This exemplifies how prior specification is a vital part of the modeling process,
and although there is no true or false prior, there are better and worse priors in terms of how well the subsequent inference will perform. In particular, the sign of the prior correlation will determine the direction in which the posterior mean is shifted in the unobserved location. Thus, if we prescribe anticorrelation ($\rho < 0$) in the prior when the actual (but usually unknown in practice) errors are correlated ($\rho > 0$) or vice-versa then the inference would not perform well for the unobserved location. Here, Tobler's first law of geography, that the behavior of nearby elements in a system will be more alike than those that are
further apart (Wikle et al., 2019), is a helpful guiding principle for spatiotemporal modeling suggesting that we should often set $\rho > 0$ if locations $i = 1$ and $i = 2$ are close in some sense. On the other hand, in some cases when the locations are further apart or $x_1$ and $x_2$ are *different* variables (e.g. one is snowdepth the other is air temperature) prescribing a prior anticorrelation $\rho < 0$ is more appropriate. In Figure A2 we show an ideal case where a strong positive prior correlation of $\rho = 0.9$ is in line



with the underlying error structure, such that the posterior inference also performs well at the unobserved location. In line
with the likelihood principle (MacKay, 2003), the prior correlation plays no role in the inference at the observed location, as
can be seen in (A9) and (A11), since there is nothing to learn from the unobserved location. This is reflected in panel (b) in
Figures A1&A2 where the marginal posterior at the observed location is identical for $\rho = 0$ and $\rho = 0.9$.

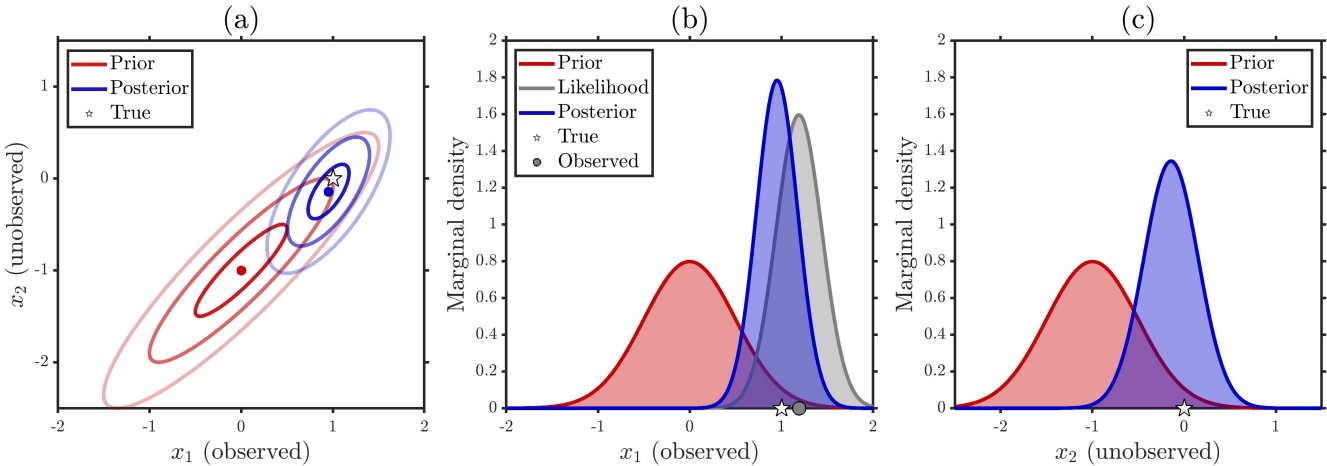

**Figure A2.** As in Figure A1 but with a correlated prior $\rho = 0.9$.

*Author contributions.* Conceptualization was by EAG and KA. Data were curated by EAG. Formal analysis was undertaken by EAG and
KA. Funding was acquired by EAG and KA. Investigation was undertaken by EAG and KA. Methodology was developed by EAG, KA,
MM and PL. Project administration was the responsibility of EAG and KA. Resources were the responsibility of NP, DT and SW. Software
was the responsibility of EAG with key contributions of KA. Supervision was carried out by NP, DT, SW, JILM and SG. Validation was
performed by EAG. Visualization was developed by EAG and KA. Writing – original draft preparation was lead by EAG and KA, with
contributions from all co-authors. Writing – review and editing was result of the common effort of all co-authors.

*Competing interests.* The contact author has declared that none of the authors has any competing interests.

*Acknowledgements.* This joint research effort was initiated through the JASPER project (no. 337515) funded by the Research Council
of Norway. Esteban Alonso-González has been funded by the CNES postdoctoral fellowship. Kristoffer Aalstad and Norbert Pirk were
funded by the Research Council of Norway through the Spot-On project (no. 301552) and acknowledge support from the LATICE Strategic
Research Initiative at the University of Oslo. Marco Mazzolini and Désirée Treichler acknowledge funding from the Research Council of



Norway (SNOWDEPTH project, contract 325519). This study was partially funded by the project SNOWDUST (TED2021-130114B-I00)
and MARGISNOW (PID2021-124220OB-100), both funded by the Spanish Ministry of Science and Innovation.



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
