# Peer review of "Spatio-temporal information propagation using sparse observations in hyper-resolution ensemble-based snow data assimilation"

_EGUsphere, 2023_

## Author Comment (AC1)

**General Comments:**

COMMENT # 1.1

> *The authors present a very interesting and promising study about spatial snow data assimilation for high-resolution simulations. The study shows how information from sparse snow depth observations can be used to improve spatially complete simulations obtained using a physically-based snow model at very high spatial resolution. The manuscript is easy to read and well-written, the results are clearly shown and discussed in depth. Overall, the paper is a strong contribution to existing literature on snow data assimilation since few studies have addressed the problem of propagating information from sites with observations to locations lacking measurements. My comments on the manuscript are only minor, and listed below.*

**Reply:**

We appreciate the reviewer's positive comments and are very grateful for the constructive suggestions that have helped us to improve this study. The following provides a point-by-point response to the reviewer's comments.

**Specific comments:**

COMMENT # 1.2

> *Abstract: The study would benefit from a shorter and more concise abstract.*

**Reply:**

Thanks for the suggestion, we have reduced the abstract by removing the first sentences as also suggested by Reviewer 2.

**Changes:**

~~Monitoring the snowpack remains challenging in part due to the limited availability of observations. On the one hand, the deployment of dense ground-based monitoring networks is hampered by logistical hurdles. On the other hand, satellite-based remote sensing products provide only partial information about the snowpack, often limited to snow-covered area or surface temperature. Numerical models are a valuable tool to help fill the gaps in snowpack monitoring. Model performance is nonetheless contingent upon the quality of meteorological forcing, which is often highly uncertain~~

 Data assimilation techniques that integrate available. . .

COMMENT # 1.3

*L 150-152: The sentence is difficult to read. Please reformulate.*

**Reply:**

We have reformulated this sentence as follows.

**Changes:**

MuSA is an open-source snow data assimilation toolbox. It is designed to assimilate  observations into simulations generated by the energy and mass balance model the Flexible Snow Model (FSM2 1), or other snow models. .

COMMENT # 1.4

*L 173: Is optimal interpolation only occasionally used in operational data assimilation? Is not many very important weather forecasting models using this method, such as ECMWF? ECMWF: IFS Documentation CY45R1 – Part II: Data assimilation, in: IFS Documentation CY45R1, IFS Documentation, ECMWF, https://www.ecmwf.int/en/elibrary/80893-ifs-documentation-cy45r1-part-ii-data-assimilation (last access: 16 November 2022), 2018.*

**Reply:**

We would like to thank the reviewer for this thoughtful comment which has made it clear to us that these OI methods are still widely used operationally, including at ECMWF. We have thus modified the text accordingly and added a more up to date reference, namely (2), on the use of OI (and other) methods in operational DA.

**Changes:**

Indeed, ensemble Kalman methods are closely related to many spatial modeling techniques, including kriging (3; 4) methods in geostatistics (5; 6) and the nearly equivalent Optimal Interpolation (7; 8) methods that  are widely used in operational DA (9; 2).

COMMENT # 1.5

*Data and methods: How were the snowpack layers in FSM2 updated during the assimilation experiments? If the assimilation step adjusts the total depth of the snowpack, also the number of modelled snow layers may change. How was this handled? Please clarify.*

**Reply:**

Unlike how data assimilation is typically implemented in the numerical weather prediction community, we do not directly update the model states in the assimilation step. Indeed, updating the snow layers of the simulation would introduce many problems, such as dealing with different numbers of layers in each ensemble member (the number of layers changes dynamically depending on the snow depth up to a maximum of 3) or the need to assume a composition of the layers (their ice and water content) at the time of the analysis, in order not to make the model unstable. Instead, we directly infer model parameters, in this case meteorological forcing correction parameters, for each water year and we let the dynamical model consistently solve for the updated state of the snowpack. This is explained on L197 and in more detail in (10). This method is not unique to our work and has been called the forcing formulation of the Bayesian data assimilation problem (11). We have now clarified this further in the text around L197.

**Changes:**

The parameters are updated with the ensemble Kalman analysis step in the transformed (Gaussian) space but fed through the forward model in the physical (untransformed) space (10). As such, we adopt a forcing formulation of the data assimilation problem (11) where model parameters are directly updated leading to indirectly but dynamically consistent updates in the model states.

COMMENT # 1.6

*Data and methods: Please specify the total computation time of running one assimilation experiment, and also add information about the computer resources that were used. This is interesting for potential applications of the methods developed in this study.*

**Reply:**

We have added the following estimation around L332.

**Changes:**

The experiments developed in this work were launched in the supercomputing facilities of the Centre National d'Études Spatiales. As a reference, 30 nodes were used, with 10 processors each. The experiments took around 5 hours, but this estimate should be taken with caution. As the operation is I/O intensive, depending on the configuration of the simulations, the computing scheme and the spatial density of observations, the computational cost can vary tremendously even for the same domain and spatial resolution.

COMMENT # 1.7

*Table 1: Would it be possible and more visually appealing to show these metrics as time series plots?*

**Reply:**

We would like to thank the reviewer for this constructive suggestion. At the same time, based on our experience with the snow data assimilation literature, it is normal to summarize the performance of experiments using multiple evaluation metrics in a table as we have done here. Generating discrete time series plots with these metrics is not straightforward given that the timing of the drone acquisitions is irregularly distributed. Such a modification would also require a relatively large figure with separate panels for each of these metrics since they vary considerably in terms of both their units and dynamic range. In summary, although a discrete time series may be more visually appealing, the use of a table such as this is a more compressed representation of the results.

**Technical comments:**

COMMENT # 1.8

*L 119: A space is missing before the reference.*

**Reply:**

Corrected.

COMMENT # 1.9

*L190: Error in spelling. "Spare" to "sparse".*

**Reply:**

Corrected.

COMMENT # 1.10

*L405: Is the sentence correct? ("which shows is nearly flat")*

**Reply:**

Thanks for spotting this typographic error, it has now been corrected.

COMMENT # 1.11

*L 518: "e.g." misplaced.*

**Reply:**

Corrected.

REFERENCES

[1] R. Essery, "A factorial snowpack model (FSM 1.0)," *Geosci. Model Dev.*, 2015.

[2] P. de Rosnay, G. Balsamo, C. Albergel, J. Muñoz-Sabater, and L. Isaksen, "Initial-isation of Land Surface Variables for Numerical Weather Prediction," *Surveys in Geophysics*, vol. 35, pp. 607–621, 2014.

[3] D. Krige, "A statistical approach to some basic mine valuation problems on the Witwatersrand," *Journal of the Southern African Institute of Mining and Metallurgy*, 1951.

[4] G. Matheron, "Principles of geostatistics," *Economic Geology*, 1963.

[5] L. Bertino, G. Evensen, and H. Wackernagel, "Sequential Data Assimilation Techniques in Oceanography," *International Statistical Review*, 2003.

[6] J. Chilès and P. Delfiner, *Geostatistics: Modeling Spatial Uncertainty*. Wiley, 2012. 2nd Edition.

[7] A. Eliassen, "Provisional report on calculation of spatial covariance and autocor-relation of the pressure field," in *Dynamic Meteorology: Data Assimilation Methods (1981)* (L. Bengtsson, M. Ghil, and E. Källén, eds.), pp. 319–330, Springer, 1954. Reprinted from Videnskaps-Akademiets Institutt for Vær-Og Klimaforskning, Oslo, Norway.

[8] L. Gandin, "Objective analysis of meteorological fields," 1963. Gridromet. Izd. Leningrad (Russian).

[9] O. Talagrand, "Assimilation of observations, an introduction," *Journal of the Meteorological Society of Japan*, pp. 191–209, 1997.

[10] E. Alonso-González, K. Aalstad, M. W. Baba, J. Revuelto, J. I. López-Moreno, J. Fiddes, R. Essery, and S. Gascoin, "The Multiple Snow Data Assimilation System (MuSA v1.0)," *Geosci. Model Dev.*, 2022.

[11] G. Evensen, F. C. Vossepoel, and P. J. van Leeuwen, "Data Assimilation Fundamentals: A Unified Formulation of the State and Parameter Estimation Problem," *Springer International Publishing*, 2022.

---

## Author Comment (AC2)

**General Comments:**

COMMENT # 1.1

*This paper shows how sparse high-resolution snow depth observations can be integrated into a hyper-resolution model that is by itself producing rather homogenous snow depths over a small catchment in the Pyrenees. The assimilated observations are a subsample taken from some drone images during the peak and melt season of snow. Three different prior (spatial, non-diagonal) error covariance matrices are constructed to use in a 3D EnKF scheme and to allow the propagation of information in space. The novelty lays in defining such error covariance matrices, using other measures than the distance between grid cells, i.e. e.g. using topography or observed (as opposed to similated) snow pack similarity. The strength of this paper is in its technical innovation and rigor, and the paper is very well written - a pleasure to read. Some details are not entirely clear and questions for clarification are listed below, together with minor suggestions to improve the presentation of the paper.*

**Reply:**

We appreciate the reviewer's positive, insightful, and constructive feedback on our work. We respond below to the comments and questions raised by the reviewer.

**Specific comments**

COMMENT # 1.2

*The ensemble generation is at the heart of this paper and 3 methodological points are a bit unclear:*

1. *The perturbations are applied (as usual) to meteorological input, but not to snow depth or any snow state variable. Do you maintain enough spread after the assimilation events this way? Most often, some extra state perturbation is desirable, either to reflect some parameter uncertainty (if the parameters were not perturbed) or just to do some covariance inflation. In addition, it could actually help to impose the right error structure in the prior state ensemble in your case – see next comment.*

2. *The prior state error correlations are technically derived from the ensemble snow depth members, which in turn were obtained by propagating ensembles of precipitation and temperature through the FSM2. Right? The newly developed error structures are thus imprinted in the forcings, not directly in the state (only after propagation through the model). Yet, the forcings are nonlinearly transformed to snow depth, and on top of that*

*the error correlations between the various forcings is assumed to not exist, meaning that the different forcing error structures will interact and might destroy or amplify each other locally. Furthermore, I did not see (may have overlooked) any temporal autocorrelation in the error structures. Therefore, I do not really understand how the resulting snow depth error correlation would have the same error structure as that of its input. Have you verified if the diagnosed ensemble state error covariance matrix (truly derived from the ensembles) effectively has the structure that it was meant to have? (From Fig 7, I get that the snow depth pattern itself (not the errors) indeed was reproduced with Exp III DA.)*

3. *Exp I, II or III: the distance, topography or similarity in snow depth are assigned to error structures in the meteorological input. Could they not equally originate from error structures in model parameters instead?*

**Reply:**

We would like to thank the reviewer for raising these three intriguing issues. We provide a point-by-point response below::

1. We do not apply any explicit inflation to the updates. Although we agree that this heuristic approach may help to improve the results by avoiding an over-confident ensemble, previous studies have noted that deterministic ensemble Kalman methods like the DES-MDA already apply an implicit inflation to the updated ensemble covariance ([1]; [2]). Moreover, a detailed evaluation of the need for explicit covariance inflation in spatio-temporal snow data assimilation would introduce another hyper-parameter requiring sensitivity analysis that is orthogonal to and thus beyond the scope of this work. Inflation is nonetheless an important and under-explored topic in the snow data assimilation literature that warrants future research. As for the direct perturbation of state variables, this would violate the strong-constraint (perfect model) assumption that is made both for convenience and consistency as discussed extensively in ([3]).

2. Yes, the prior covariances in the state variables are generated by propagating the spatially correlated parameter ensemble through the forward model which is FSM2 in this case. So, given our forward model, there is thus a non-linear relationship between the forcing perturbation parameters that we update directly and the predicted snow depth observations. This non-linearity is the major reason that we resort to using iterative ensemble Kalman methods. Since the advantage of iterative methods was not discussed in the original manuscript we have now added an explanation and relevant references to section 2.2.2.

For simplicity (see line 209) we do not assume any prior correlation between the respective types of perturbation parameters. Although it would be possible and maybe advantageous to do this (e.g. (4)), here we do not have a strong prior belief that would justify any particular such across parameter correlation. It is hoped that a good posterior covariance between these parameters can be inferred from the data, provided that the prior spatial correlation models are reasonable. Along with assimilating different observations (5), this across parameter prior correlation is a promising way to help address equifinality related to the compensating effects in melt and accumulation parameters (e.g. (6)) that the reviewer is possibly alluding to, but that is beyond the scope of this work. As for the temporal auto-correlation in the ensemble of states, this is also set implicitly through the perturbation parameters and their propagation through the forward model. This temporal auto-correlation is thus likely high given that the forcing perturbation parameters are time-invariant within each water year.

When it comes to the ensemble covariance matrices having the right structure, we assume that this question only applies to Exp. III where we used snow depth observations to design the prior parameter correlation model. In the other experiments, we do not expect anything a-priori other than there being some specified geographic distance (Exp. I) and topographic distance-based (Exp II) correlation (defined by the GC function) in the perturbation parameters that we update. It is important to note that in all experiments we do not try to define some kind of true or correct prior (in that all priors are subjective (7; 8)) instead we hope to design priors that allow for improved spatio-temporal information propagation across grid cells so as to obtain better estimates of snow depth. In Exp. III we inherit some of the spatial correlations in the snow depth observations when designing the prior parameter correlation model. We do not expect the spatial correlations between snow depths and perturbation parameters to be the same, given the non-linear forward model, but we hope these correlations can help improve the performance of the spatio-temporal data assimilation. The encouraging results from Exp. III suggest empirically that this is indeed the case. Indeed, we did not actually expect Exp III to perform as well as it did given the aforementioned non-linearity. We apologize if we have misunderstood this astute technical question from the reviewer, but we hope that this response has nonetheless addressed some of the reviewer's concerns.

3. Yes, it is of course a possible option. The way we correct the forcing is by generating perturbation parameters that we use to correct the meteorological time-series. It is also possible (and it is supported by MuSA) to include internal model parameters within the analysis. However, the available literature, such

as (9), and our own experience implies that forcing is almost always the biggest source of uncertainty in snow simulations. Furthermore, it is a common practice in the snow data assimilation literature to include only forcing correction parameters (see (3) and references therein). For these reasons, and in the interest of simplicity, we have focused only on the forcing correction.

**Changes:**

The iterative ensemble Kalman methods used herein could nonetheless readily be extended to filtering (1; 10). Although the iterations incur an additional computational cost, they allow for likelihood tempering (11) that leads to improved performance compared to non-iterative methods when the model mapping from parameters to observations is non-linear (12; 3; 4; 13). The snow data assimilation problem addressed herein falls under this non-linear category.

COMMENT # 1.3

*Perturbing precipitation w/ a logit-normal distribution and a mean of -1.6 seems to introduce a bias in precipitation. Is there a reason for this choice?*

**Reply:**

We would like to thank the reviewer for raising this point which we have now clarified in the text. The prior hyperparamters $\mu_0$ and $\sigma_0$ of the weakly informative logit-normal prior distributions were selected based on initial prior predictive checks (not shown) based in part on previous experiments and ensuring a reasonable spread in the ensemble without having unphysical seasonal snowpacks (e.g. glaciers). It is worth noting that $-1.6$ is not the mean of the logit-normal distribution itself, but the mean of the underlying normal distribution in transformed (unbounded) space. With bounds $(0, 8)$ for the precipitation perturbation, this corresponds to a right-skew logit-normal distribution with a median of 1.34 and an interquartile range of approximately $(0.74, 2.3)$ in physical (i.e., model) space. So this prior is thus not strongly biasing the typical (i.e. central) values of the precipitation perturbation parameter. We have now clarified this in the text around L200.

**Changes:**

The mean $\mu_0$ and standard deviation $\sigma_0$ of the underlying normal distributions hyper-parameters in transformed space, control the shape of our weakly informative prior probability distributions (8). These hyper-parameters were selected based on prior predictive checks and conservative expectations of the range of uncertainty in the meteorological forcing, which we distilled from experience obtained in previous studies at Izas. For temperature, the prior additive perturbation parameters were

drawn from a logit-normal distribution bounded between $-8$ and $8$ K, with hyper-parameters $\mu_0 = 0$ and $\sigma_0 = 0.5$. In physical space this corresponds to a symmetric logit-normal distribution with a median (interquartile range) of 0 ($-1.34$ to $1.34$). The prior multiplicative perturbation parameters for precipitation were drawn from a logit-normal distribution bounded between $0$ and $8$ with with $\mu_0 = -1.6$ and $\sigma_0 = 1$. In physical space this corresponds to a right-skew logit-normal distribution with a median (interquartile range) of 1.34 (0.75 to 2.27). The number of ensemble members was fixed at $N_e = 100$ for all experiments.

**COMMENT # 1.4**

*FSM2 is run with MicroMet data. Can you explicitly say how much variability you expect at the scale of this small catchment? I would expect almost none, perhaps truly nothing at all for precipitation, but I wonder if there is any (other than some radiation variability mentioned in the results). Is there a reason why the wind distribution is not turned on?*

**Reply:**

As the reviewer correctly points out, since wind redistribution is not considered most of the snowmelt variability is induced by radiation (even at coarser resolutions), at least in these kind of high mountain Mediterranean environments (14). This is enough to induce some variability in the SWE patterns (Fig. 3), although compared to posterior simulations greatly underestimated (See Animation 1 for an example). The wind distribution is not enabled as it is not part of MicroMet, it is part of another module (SnowTran) in the Snowmodel package. Thus, it is not implemented in MuSA which runs FSM2 at its core.

**COMMENT # 1.5**

*The text mentions both ensemble Kalman filter and smoother (e.g. L. 244 and further). The time dimension (smoothing) is unclear to me, if it is applied. Which exact technique is applied?*

**Reply:**

We have now clarified this in section **2.2.2** as follows

**Changes:**

To perform the spatio-temporal data assimilation we employed the deterministic ensemble smoother with multiple data assimilation (DES-MDA) scheme introduced by (2).  We use this batch smoother scheme to directly update parameters (and

indirectly update states) by simultaneously assimilating all available observations in the data assimilation window as outlined in Algorithm 1. Herein, the data assimilation window is one water year and the parameters are updated independently for each water year. This DES-MDA is a deterministic version of the original...

COMMENT # 1.6

*Not sure about the technical details for the computational implementation: could you not use halos around a good radius of influence to parallelize in space as is done in 3D global DA systems?*

**Reply:**

MuSA has been designed to run for so-called embarrassingly parallel models without lateral interactions. That is, the model prediction for each grid cell is simulated independently. This is the reason why the implementation is challenging as its possible to de-synchronise each iteration of the algorithm (as explained in section 2.4 Computational setup), since each processor generates an ensemble of simulations for each cell. Once all ensembles are created and stored as individual files, during the analysis of each individual cell, MuSA searches for non-local ensembles containing observations to build the matrices involved in the Kalman step. The implementation is therefore similar to that described by the reviewer using halos (domain localization), but there is also the need to control the solution of cells at different nodes so as not to de-synchronise the simulations. One advantage of this is that at the analysis of each cell, only the non-local cells with observations have to be read, instead of files storing distributed simulations. Moreover, in principle any embarrassingly parallel model running at the single cell level, including many LSMs or more complex snow models (or alternatively simpler and therefore more efficient), will be compatible with our implementation.

COMMENT # 1.7

*Evaluation: it is a pity that there are no in situ data available of any sort, but I agree that your setup does its job for the application at hand. Would be nice to try to assimilate e.g. lidar (e.g. ASO?) or radar (Sentinel1?) data and have drone data as reference data.*

**Reply:**

We thank these are great suggestions. We agree that in the future it would be instructive to explore the use of the drone dataset as ground truth in experiments assimilating other retrievals. In this work our primary goal was to explore how to propagate information in these hyper-resolution simulations, a technique that we hope will have broad implications. Continued research and exploration of more generalizable experiments using emerging snow remote sensing data, such as the ICESat-2 laser altimiter as mentioned on L519, should be encouraged and is a topic that we aim to pursue further.

**Technical comments:**

COMMENT # 1.8

*The abstract is long, and the first 7 lines can be removed. This is a rather technical paper, and there is no need for an extensive introduction in the abstract.*

**Reply:**

Following the reviewer's suggestion, we have removed the first 7 lines of the abstract.

**Changes:**

~~Monitoring the snowpack remains challenging in part due to the limited availability of observations. On the one hand, the deployment of dense ground-based monitoring networks is hampered by logistical hurdles. On the other hand, satellite-based remote sensing products provide only partial information about the snowpack, often limited to snow-covered area or surface temperature. Numerical models are a valuable tool to help fill the gaps in snowpack monitoring. Model performance is nonetheless contingent upon the quality of meteorological forcing, which is often highly uncertain especially in complex terrain. To address these limitations, data~~ Data assimilation techniques that integrate available...

COMMENT # 1.9

*L. 119: space before reference*

**Reply:**

Corrected.

COMMENT # 1.10

*L. 215: should $d$ be $d_ij$ in line w/ Eq. 7, and should $d_ij$ on L.223 be redefined as some $D$ matrix? I think $d_ij$ refers to a distance for a single pair of grid cells.*

**Reply:**

Strictly speaking the distance in the function in Eq. (1) is continuous rather than discrete. The only reason it later becomes discrete is because we are running a spatially discretized model with a finite-dimensional state space. We nonetheless agree that our notation here was somewhat sloppy and have rectified it, especially on L223, as outlined below. In particular, the reviewer is correct that $d_{ij}$ refers to the (generalized) distance between two grid cells and that this is an element of a larger (square symmetric) distance matrix $\mathbf{D}$.

**Changes:**

In DA, the continuous distance $d$ in the GC function is typically  discretized to a Euclidean distance between two spatial grid cells defined in two (easting and northing) or three (with elevation) dimensional geographic space. The concept can be generalized, however, to be any measure of distance between two grid cells as detailed in Section 2.3. In our experiments, we have selected two different values of the hyperparameter $c$ after manual tuning. For Experiment I $c = 100$, while for Experiments II and III $c = 5$ (Section 2.3). The difference in the magnitude of the $c$ value in the different experiments is a consequence of the covariance-based normalization of the distance matrix ( $\mathbf{D} = [d_{ij}]$) in Experiments II and III.

COMMENT # 1.11

*L. 296: I agree that it is fine to do this; we also call that the use of statistical "signatures".*

**Reply:**

We would like to thank the reviewer for endorsing the validity of this methodology. We were not aware that the term statistical signature was used in this context, but we will now keep a look out for connections with this approach in the future.

COMMENT # 1.12

*Algorithm 1 box - line 13 bis: Y(i) is not defined, should it just be y(i)?*

**Reply:**

Thanks for spotting this, we had indeed forgotten to properly define $\mathbf{Y}^{(i)}$. This matrix is introduced for dimensional consistency and it is simply a $N_o^{(i)} \times N_e$ matrix containing $N_e$ copies of the local (i.e. $d < 2c$) observation vector $\mathbf{y}^{(i)}$. We have now included the definition $\mathbf{Y}^{(i)} = \mathbf{y}^{(i)} \mathbf{1}_{N_e}^T$ involving an outer product on line 13 in the Algorithm 1.

COMMENT # 1.13

L. 411: typo "horizontal"

**Reply:**

Corrected.

COMMENT # 1.14

*Fig 8: units of snow volume are right for an area of 55 ha, but I would just write them in average snow depth [m].*

**Reply:**

We are of the opinion that showing the results in units of volume is more instructive to show how well we are able to constrain the total amount of snow in the catchment using just a few sparse observations. Since this is admittedly a subjective decision and some readers (such as the reviewer) may prefer to think of this in terms of average snow depth, we have now included the relevant scaling factor to convert to average snow depth (in meters) in the caption of Figure 8.

**Changes:**

Time evolution of the total volume of snow in the catchment in units of cubic hectometers ($10^6$ $m^3$) from the different simulations (colored lines) along with the volume estimates obtained from the snow depth drone maps (blue dots). Note that the snow volume can be multiplied by a scale factor of 1.82 to recover the mean snow depth for the Izas catchment with an area of 55 hectares.

**REFERENCES**

[1] P. Sakov and P. Oke, " A deterministic formulation of the ensemble Kalman filter: an alternative to ensemble square root filters," *Tellus A: Dynamic Meteorology and Oceanography*, 2008.

[2] A. Emerick, "Deterministic ensemble smoother with multiple data assimilation as an alternative for history-matching seismic data," *Computational Geosciences*, 2018.

[3] E. Alonso-González, K. Aalstad, M. W. Baba, J. Revuelto, J. I. López-Moreno, J. Fiddes, R. Essery, and S. Gascoin, "The Multiple Snow Data Assimilation System (MuSA v1.0)," *Geosci. Model Dev.*, 2022.

[4] N. Pirk, K. Aalstad, S. Westermann, A. Vatne, A. van Hove, L. Tallaksen, M. Cassiani, and G. Katul, "Inferring surface energy fluxes using drone data assimilation in large eddy simulations," *Atmospheric Measurement Techniques*, 2022.

[5] E. Alonso-González, S. Gascoin, S. Arioli, and G. Picard, "Improving numerical snowpack simulations by assimilating land surface temperature," *EGUsphere*, 2022.

[6] K. Aalstad, S. Westermann, T. V. Schuler, J. Boike, and L. Bertino, "Ensemble-based assimilation of fractional snow-covered area satellite retrievals to estimate the snow distribution at Arctic sites," *The Cryosphere*, 2018.

[7] D. J. C. MacKay, *Information Theory, Inference, and Learning Algorithms*. Cambridge University Press, 2003.

[8] K. Banner, K. Irvine, and T. Rodhouse, "The use of Bayesian priors in Ecology: The good, the bad and the not great," *Methods Ecol. Evol.*, 2020.

[9] M. S. Raleigh, B. Livneh, K. Lapo, and J. D. Lundquist, "How does availability of meteorological forcing data impact physically based snowpack simulations?," *J. Hydrometeorol.*, 2016.

[10] A. A. Emerick and A. C. Reynolds, "History matching time-lapse seismic data using the ensemble Kalman filter with multiple data assimilations," *Computational Geosciences*, 2012.

[11] A. Stordal and A. Elsheikh, "Iterative ensemble smoothers in the annealed importance sampling framework," *Advances in Water Resources*, vol. 86, pp. 231–239, 2015.

[12] A. A. Emerick and A. C. Reynolds, "Ensemble smoother with multiple data assimilation," *Computers & Geosciences*, 2013.

[13] G. Evensen, F. C. Vossepoel, and P. J. van Leeuwen, "Data Assimilation Fundamentals: A Unified Formulation of the State and Parameter Estimation Problem," *Springer International Publishing*, 2022.

[14] M. Baba, S. Gascoin, C. Kinnard, A. Marchane, and L. Hanich, "Effect of digital elevation model resolution on the simulation of the snow cover evolution in the High Atlas," *Water Resources Research*, vol. 55, pp. 5360–5378, 2019.